# Reliable Confidence Alignment for Generalized Category Discovery

**Jiawei Yu** [1 2]  **Zijian Gao** [1 2]  **Tianjiao Wan** [1 2]  **Xuan Liu** [3]  **Cheng Yang** [1 2]  **Kele Xu** [1 2]

## Abstract

Generalized Category Discovery (GCD) requires models to categorize an unlabeled pool containing known and novel classes under sparse supervision. We identify a systemic confidence bias inherent in existing parametric methods: while entropy regularization prevents class collapse, it indiscriminately suppresses predictive certainty on all unlabeled instances. This bias drives a distributional wedge between labeled and unlabeled samples of the same category, forcing models to sacrifice their stability on known classes to achieve plasticity for new ones. To resolve this, we propose Reliable Confidence Alignment (RCA), a plug-and-play framework grounded in Evidential Deep Learning. RCA first establishes high certainty anchors on labeled data using a Reliable Anchor for Certainty (RAC) module. Then, we introduce Cross-view Confidence Alignment (CCA) to propagate this grounded reliability to the unlabeled discovery set. Thus, RCA captures the fine-grained geometry of the probability simplex, effectively calibrating the model's epistemic uncertainty. Extensive evaluations on coarse- and fine-grained benchmarks demonstrate that RCA effectively rectifies the confidence landscape, significantly mitigating performance decay on known classes without compromising novel-class discovery.

## 1. Introduction

The practical utility of traditional deep models is largely tied to the closed-world assumption, a framework where training and deployment occur within a fixed taxonomy (Zhou, 2022; Yang et al., 2024). While this premise allows for

*Corresponding author. [1]College of Computer Science and Technology, National University of Defense Technology [2]State Key Laboratory of Complex & Critical Software Environment [3]College of Computer Science and Electronic Engineering, Hunan University. Correspondence to: Kele Xu <xukelele@nudt.edu.cn>.

*Proceedings of the 43[rd] International Conference on Machine Learning*, Seoul, South Korea. PMLR 306, 2026. Copyright 2026 by the author(s).

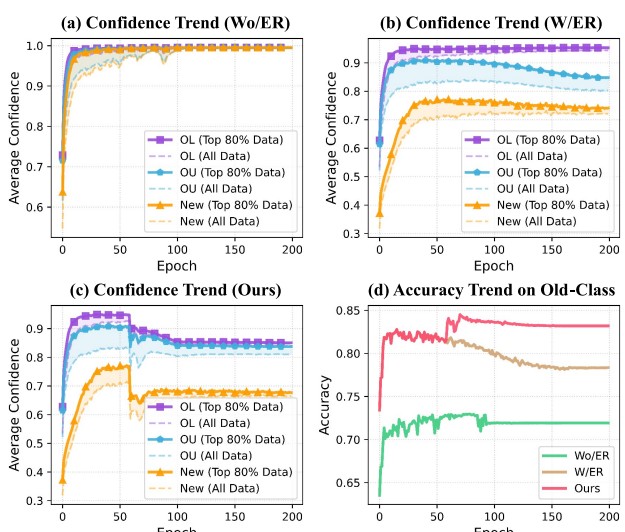

*Figure 1.* Prediction confidence trends on old-class labeled (OL), old-class unlabeled (OU), and new-class (New) data, and old-class performance under different strategies. Wo/ER and W/ER denote methods without and with entropy regularization, respectively.

high performance in controlled settings, it falls short in real-world applications such as autonomous driving (Bai et al., 2024; Wang et al., 2022) or medical diagnostics (Yue et al., 2025), where novel categories emerge unexpectedly. This has led to the emergence of Generalized Category Discovery (GCD) (Vaze et al., 2022), a paradigm requiring models to classify an unlabeled data pool that contains a mixture of previously seen and entirely novel categories. Unlike earlier tasks such as Novel Category Discovery (Han et al., 2019; 2021; Fini et al., 2021), which assume the labeled and unlabeled sets have disjoint classes, GCD operates under the more realistic condition that unlabeled data includes instances from both known categories and unseen ones, thereby constituting a more challenging scenario.

In contemporary research, parametric discovery methods (Ma et al., 2025; Xu et al., 2025) have emerged as a prevalent paradigm due to their superior inference efficiency compared to clustering-based approaches. These frameworks typically leverage self-distillation (Caron et al., 2021) and contrastive learning (Chen et al., 2020; Khosla et al., 2020) for category discovery, while widely adopting entropy maximization regularization (Wen et al., 2023) to prevent the model from blindly overfitting to known (old) classes (Figure 1(a)), thereby avoiding suboptimal perfor-

mance (Green curve in Figure 1(d)). Although this regularization effectively circumvents prediction collapse, it triggers a systemic **stability–plasticity dilemma**.

As shown in Figure 1(b), entropy regularization creates a systematic confidence mismatch between labeled and unlabeled samples belonging to the same old class. Consequently, many unlabeled old-class instances are pushed into low-confidence regions that are primarily associated with novel categories, blurring decision boundaries and degrading performance on known classes in an imbalanced pursuit of plasticity (Brown curve in Figure 1(d)). We term this phenomenon **Confidence bias**.

To address this misalignment, we propose **Reliable Confidence Alignment (RCA)**, a plug-and-play framework that explicitly aligns epistemic confidence across labeled and unlabeled data. Instead of operating on point-estimate probabilities, RCA reformulates category discovery from a distribution-aware perspective using Evidential Deep Learning (EDL). RCA consists of two complementary components. Reliable Anchor for Certainty (RAC) establishes reliable confidence anchors on labeled data by modeling class predictions as evidence-supported distributions, thereby counteracting the confidence suppression induced by entropy regularization. Building upon these anchors, Cross-view Confidence Alignment (CCA) propagates grounded reliability to unlabeled samples by aligning Dirichlet distributions across different views, enabling consistent uncertainty structure beyond final class probabilities. Our main contributions are as follows:

(1) We identify confidence bias in entropy-regularized GCD as a root cause of the stability–plasticity dilemma. To mitigate this issue, we propose RCA, a plug-and-play framework that effectively achieves confidence alignment.

(2) We propose RAC, which leverages the labeled set to enable Dirichlet distribution-aware certainty anchoring for confidence compensation.

(3) We introduce CCA to align confidence structures across different views by maintaining first- and second-order consistency across the probability simplex, effectively capturing epistemic uncertainty.

(4) Extensive experiments on benchmarks of varying granularities demonstrate that RCA consistently improves old-class performance while preserving robust novel-class discovery across both weak and strong GCD baselines.

**Conflict of Interest Disclosure** We declare that we have no financial conflicts of interest to disclose.

## 2. Related Work

**Generalized Category Discovery (GCD).** Novel Category Discovery (NCD) (Chi et al., 2022; Liu et al., 2023; Li et al., 2023) seeks to transfer knowledge from labeled classes to classify unlabeled samples from unseen categories only. GCD further broadens the scope of NCD by relaxing the assumption that the unlabeled pool consists solely of novel categories, thereby constituting a more realistic setting. Vaze et al. (Vaze et al., 2022) establish a benchmark for this task and demonstrate that a simple non-parametric classifier built upon strong self-supervised representations can serve as a powerful baseline. SimGCD (Wen et al., 2023) attributes previous parametric failures to overconfidence on known classes and introduces entropy regularization, forming a strong parametric baseline. Recent state-of-the-art methods largely build upon the SimGCD paradigm. SPTNet (Wang et al., 2024a) employs spatial prompt tuning to focus on different regions of an image, particularly foreground object regions. ProtoGCD (Ma et al., 2025) achieves unified modeling of known and novel classes by jointly leveraging prototypes and a unified learning objective. AllGCD (Cao et al., 2025) leverages intra-class contrast on labeled data and inter-class contrast on unlabeled data to jointly promote intra-class compactness and inter-class separation. Despite their success, most of the methods typically apply uniform entropy regularization to both known and novel classes, which inherently leads to a trade-off between their respective performance during training.

**Evidential Deep Learning (EDL).** EDL is commonly used for uncertainty estimation. Unlike approaches that directly predict class probabilities via the softmax function, EDL is grounded in Dempster–Shafer Theory (DST) (Shafer, 1992) and quantifies the "evidence" supporting each class by transforming network outputs into the parameters of a Dirichlet distribution, which serves as the conjugate prior of the multinomial distribution. Compared to Bayesian neural networks that require costly MCMC or variational inference, EDL captures epistemic uncertainty with significantly lower computational overhead. Sensoy et al. (Sensoy et al., 2018) were the first to combine Subjective Logic (SL) (Jøsang, 2016) with DST, significantly improving model robustness. Existing studies on EDL extensions have predominantly focused on supervised or semi-supervised learning scenarios. In contrast, we are the first to introduce EDL into the open-world setting of GCD.

## 3. Preliminary Analysis

### 3.1. Problem Definition

GCD aims to exploit labeled data from known classes to simultaneously recognize known categories and discover novel ones from unlabeled data. Formally: given a labeled dataset $\mathcal{D}_l = \{(x_i^l, y_i^l)\} \in \mathcal{X} \times \mathcal{Y}_l$ and an unlabeled dataset $\mathcal{D}_u = \{(x_i^u)\} \in \mathcal{X}$ with label space $\mathcal{Y}_u$, where $\mathcal{Y}_l \subset \mathcal{Y}_u$. The total number of classes that GCD needs to distinguish is $K = K_{\text{old}} + K_{\text{new}}$, where $K_{\text{old}} = |\mathcal{Y}_l|$. $K_{\text{new}}$ can be

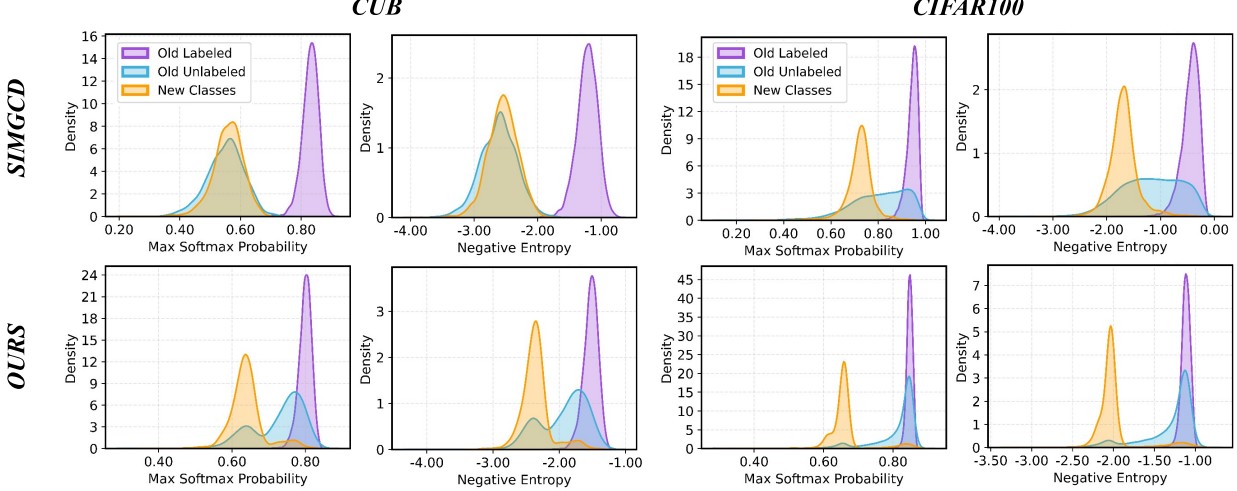

*Figure 2.* Empirical diagnosis of the confidence bias via predictive distribution analysis on SIMGCD (Wen et al., 2023) baseline.

either predefined or estimated using existing methods (Han et al., 2021). We adopt the former approach to maintain consistency with previous works and to better analyze the potential problems of GCD.

### 3.2. Empirical Analysis of Confidence Bias

As shown in Figure 1(d), parametric GCD frameworks often face a persistent optimization hurdle: as the model begins to cluster novel categories, its accuracy on known (old) classes declines significantly during the later stages of training. This behavior manifests the stability–plasticity dilemma, where pursuing categorical plasticity for new groups erodes the stability of the model's existing knowledge.

In the Figure 2 and Figure 7, empirical analysis of the model's confidence landscape reveals a pronounced discrepancy between labeled old-class instances and their unlabeled counterparts. Standard entropy regularization—widely adopted by methods like SimGCD to prevent class collapse—forces the model to maximize mean predictive entropy. While this prevents the model from ignoring novelty, it creates a side effect: it indiscriminately suppresses prediction certainty across the entire unlabeled set.

This strategy overlooks a fundamental property: labeled and unlabeled old-class samples are drawn from the same underlying distribution and should ideally reside within the same high-confidence manifolds. Instead, current regularization forces unlabeled known-class samples into low-confidence regions typically reserved for novel categories. This distributional shift induces a systemic Confidence Bias, confounding decision boundaries and causing the model to misclassify familiar concepts as new clusters. This suggests that existing parametric approaches often achieve discovery by sacrificing the reliability of the established taxonomy rather than through true generalization.

## 4. Methodology

As shown in Figure 3, the proposed framework consists of two synergistic components. We first establish reliable certainty anchoring via RAC, which performs the initial alignment by grounding predictive confidence on labeled old-class data. Building upon this anchor, CCA transfers the calibrated reliability to unlabeled data, enabling global confidence alignment across diverse data types in GCD.

### 4.1. Reliable Anchor for Certainty (RAC)

To address the confidence bias issue identified in Section 3.2, we use labeled samples as reliable certainty anchors to align the confidence intervals between the two types of old-class data. Based on Dempster–Shafer Theory, RAC can be described as optimizing the combination of prediction risk and KL-divergence regularization.

#### 4.1.1. THE DIRICHLET GENERATIVE PROCESS

In order to model the distribution of old-class probabilities within the GCD framework, we introduce a Dirichlet generative process. This process treats the probabilities $\boldsymbol{\mu} = [\mu_1, \ldots, \mu_{K_{old}}]^\top$ as latent variables governed by a Dirichlet distribution, as formally defined in Definition 4.1.

**Definition 4.1 (Dirichlet Distribution).** Consider the probability simplex $\Delta^{K_{old}-1}$ endowed with its canonical measure. The Dirichlet distribution defines a probability measure over $\Delta^{K_{old}-1}$, parameterized by a vector of concentration parameters $\boldsymbol{\alpha} \in \mathbb{R}_+^{K_{old}}$. Its probability density function is given by:

$$\mathrm{Dir}(\boldsymbol{\mu} \mid \boldsymbol{\alpha}) = \begin{cases} \frac{1}{B(\boldsymbol{\alpha})} \prod_{k=1}^{K_{old}} \mu_k^{\alpha_k - 1} & \text{for } \boldsymbol{\mu} \in \Delta^{K_{old}-1}, \\ 0 & \text{otherwise}, \end{cases}$$

(1)

where $B(\boldsymbol{\alpha})$ denotes the multivariate Beta function, and the

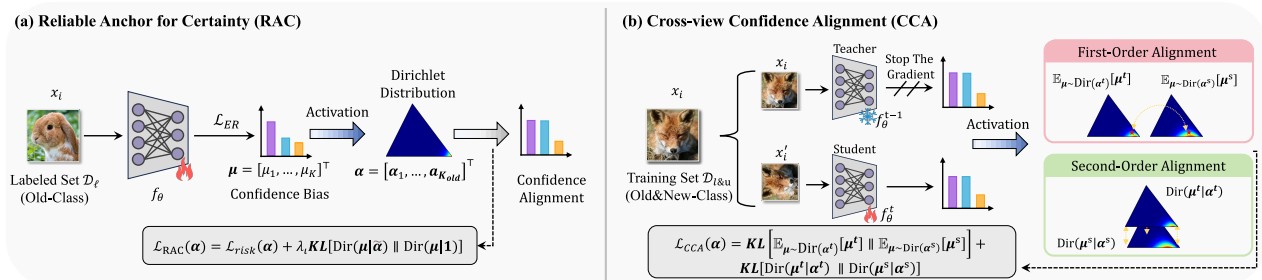

*Figure 3.* **Overview of the Reliable Confidence Alignment (RCA) framework.** The framework integrates two synergistic modules: (a) RAC grounds the evidential network on labeled data to establish high-fidelity certainty anchors, counteracting entropy-induced confidence suppression. (b) CCA propagates grounded reliability to the unlabeled discovery set by enforcing first- and second-order consistency on the probability simplex, effectively aligning Dirichlet distributions across diverse stochastic views.

$(K_{\text{old}} - 1)$-dimensional unit simplex is defined as:

$$\Delta^{K_{\text{old}}-1} = \left\{ \boldsymbol{\mu} \in \mathbb{R}^{K_{\text{old}}} \mid \sum_{k=1}^{K_{\text{old}}} \mu_k = 1, \ \mu_k \geq 0 \right\}. \quad (2)$$

Beyond serving as a standard discriminative classifier, we re-purpose the model $f_\theta$ as an evidence network. Formally, the network parameterizes the concentration vector $\boldsymbol{\alpha} = \mathbf{e} + 1$ of the Dirichlet posterior, where $\mathbf{e} = \sigma(f_\theta(\mathbf{x}))$ denotes the non-negative evidence vector induced via an activation function $\sigma(\cdot)$. We choose $\sigma = \text{Exp}(\text{Tanh}(\cdot))$ as the method for constructing the evidence. The total evidence is then denoted as $S = \sum_{k=1}^{K_{\text{old}}} \alpha_k$. Following Subjective Logic, the predictive uncertainty is quantified as $u = K_{\text{old}}/S$, representing the vacancy of belief mass in the simplex $\Delta^{K_{\text{old}}-1}$.

### 4.1.2. VARIATIONAL LOWER BOUND FOR EVIDENCE ACQUISITION

To train the evidential network $f_\theta$, we aim to maximize the marginal likelihood (model evidence) of the ground truth labels, $P(\mathbf{y}|\mathbf{x})$. Following the variational inference framework, we introduce a variational posterior $q_\theta(\boldsymbol{\mu}|\mathbf{x}) = \text{Dir}(\boldsymbol{\mu}|\boldsymbol{\alpha})$ to approximate the true posterior.

The log-marginal likelihood can be decomposed as follows:

$$\log p(\mathbf{y}|\mathbf{x}) = \mathcal{L}_{\text{ELBO}}(\theta) + \text{KL}\left[q_\theta(\boldsymbol{\mu}|\mathbf{x})\|p(\boldsymbol{\mu}|\mathbf{x}, \mathbf{y})\right], \quad (3)$$

where $\text{KL}[\cdot\|\cdot]$ denotes the Kullback-Leibler divergence. Since the KL divergence is non-negative, $\mathcal{L}_{\text{ELBO}}(\theta)$ constitutes a valid lower bound for the model evidence:

$$\mathcal{L}_{\text{ELBO}}(\theta) = \mathbb{E}_{q_\theta(\boldsymbol{\mu}|\mathbf{x})}\left[\log p(\mathbf{y}|\boldsymbol{\mu})\right] - \text{KL}\left[q_\theta(\boldsymbol{\mu}|\mathbf{x})\|p(\boldsymbol{\mu}|\mathbf{x})\right]. \quad (4)$$

Maximizing this lower bound equals minimizing the negative ELBO. Thus, we arrive at our learning objective:

**Theorem 4.2.** *The optimization of the evidential network is equivalent to minimizing the Bayesian Risk, defined by the* *expected cross-entropy and a KL-divergence regularizer:*

$$\mathcal{L}(\theta) = -\mathcal{L}_{ELBO}(\theta) \approx \underbrace{\mathbb{E}_{\boldsymbol{\mu}\sim Dir(\boldsymbol{\alpha})}[-\log p(\mathbf{y}|\boldsymbol{\mu})]}_{\mathcal{L}_{risk}}$$
$$+ \underbrace{KL\left[Dir(\boldsymbol{\mu}|\boldsymbol{\alpha})\|Dir(\boldsymbol{\mu}|\mathbf{1})\right]}_{\mathcal{L}_{reg}}. \quad (5)$$

*where $\mathcal{L}_{risk}$ represents the expected prediction risk and $\mathcal{L}_{reg}$ serves as the probabilistic regularization term matching the variational posterior to the prior.*

(*Proof.* See the Appendix A.1- A.3)

**The Expected Risk.** The first term in Eq. (5) corresponds to the expected prediction error under the induced Dirichlet posterior. For a labeled known class sample with a one-hot target $\mathbf{y}_i$, this term can be solved analytically using the properties of the Dirichlet distribution:

$$\mathcal{L}_{\text{risk}}(\boldsymbol{\alpha}_i) = \sum_{k=1}^{K} y_{ik}\left(\psi(S_i) - \psi(\alpha_{ik})\right), \quad (6)$$

where $\psi(\cdot)$ is the digamma function. This loss forces the model to accumulate evidence (i.e., $\alpha_{ik} \uparrow$) for the ground-truth class, thereby reducing the predictive uncertainty $u_i$. (*Derivation.* See the Appendix A.4)

**The KL-Divergence Constraint.** In GCD, the early stages of training often exhibit overconfidence in known classes (Wen et al., 2023). Minimizing $\mathcal{L}_{\text{risk}}$ alone can lead to the generation of misleading evidence for incorrect classes (i.e., $\alpha_j \gg 1$ for $j \neq y$). To penalize this, we impose a constraint ensuring that the posterior Dirichlet distribution diverges from the uniform prior (zero evidence state) *only* for the correct class.

We construct a target Dirichlet distribution $\tilde{\boldsymbol{\alpha}}_i = \mathbf{y}_i + (1 - \mathbf{y}_i) \odot \boldsymbol{\alpha}_i$. This formulation implies that for incorrect classes, we wish the evidence to revert to the prior parameters ($\boldsymbol{\alpha} \rightarrow \mathbf{1}$), while for the correct class, we accept

the learned evidence. The closed-form loss is derived as:

$$
\begin{aligned}
\mathcal{L}_{\text{reg}}(\boldsymbol{\alpha}_i) &= \lambda_t \text{KL}\left[\text{Dir}(\boldsymbol{\mu}|\tilde{\boldsymbol{\alpha}}_i)\|\text{Dir}(\boldsymbol{\mu}|\mathbf{1})\right] \\
&= \log\left(\frac{\Gamma(\sum_{k=1}^{K}\tilde{\alpha}_{ik})}{\Gamma(K)\prod_{k=1}^{K}\Gamma(\tilde{\alpha}_{ik})}\right) \\
&\quad + \sum_{k=1}^{K}(\tilde{\alpha}_{ik}-1)\left[\psi(\tilde{\alpha}_{ik})-\psi\left(\sum_{j=1}^{K}\tilde{\alpha}_{ij}\right)\right].
\end{aligned}
\tag{7}
$$

Here, $\Gamma(\cdot)$ denote the Gamma function. $\lambda_t = \min(1, 2t/T_{\text{anneal}})$ acts as an annealing coefficient, allowing the model to explore the parameter space freely in early epochs before imposing strict probabilistic regularization. $T_{\text{anneal}}$ is the total number of training epochs for RAC.

**For the RAC Component**: The holistic objective on the labeled set $\mathcal{D}_l$, which integrates the expected risk with KL-Divergence constraint, is given by:

$$
\mathcal{L}_{\text{RAC}} = \frac{1}{|\mathcal{D}_l|}\sum_{\mathbf{x}_i \in \mathcal{D}_l}\left(\mathcal{L}_{\text{risk}}(\boldsymbol{\alpha}_i) + \mathcal{L}_{\text{reg}}(\boldsymbol{\alpha}_i)\right).
\tag{8}
$$

### 4.2. Cross-view Confidence Alignment (CCA)

To extend the robust compensation for biased confidence established by RAC to the unlabeled set, we propose Cross-view Confidence Alignment. Grounded in PAC-Bayesian theory (Germain et al., 2009; Xiang et al., 2025), CCA employs the Dirichlet distribution as a bridge to facilitate the transfer of uncertainty knowledge at both macro and micro levels. CCA thereby addresses the uncertainty overlooked by conventional GCD methods during self-distillation.

#### 4.2.1. Theoretical Analysis of CCA

Let $\mathcal{D}$ be the underlying data distribution and $\mathcal{D}_{l\&u}$ be the training set of size $N$. Within the PAC-Bayesian theory, we define the student's expected misclassification risk relative to the teacher as $R^s = \mathbb{E}_{\mathbf{x}\sim\mathcal{D}}[-\bar{\boldsymbol{p}}^t \log \bar{\boldsymbol{p}}^s]$. The corresponding empirical risk is $\hat{R}^s = \frac{1}{N}\sum_{i=1}^{N}[-\bar{\boldsymbol{p}}_i^t \log \bar{\boldsymbol{p}}_i^s]$.

**Theorem 4.3.** *For any $\delta \in (0,1]$ and $\gamma \in \mathbb{R}^+$, the expected risk $R^s$ is bounded with probability at least $1-\delta$:*

$$
R^s \le \hat{R}^s + \frac{\gamma}{N}\sum_{i=1}^{N}KL\left(Dir(\boldsymbol{\alpha}_i^t) \| Dir(\boldsymbol{\alpha}_i^s)\right) + C(\delta),
\tag{9}
$$

*where $C(\delta)$ is a constant.*

*(Proof. See the Appendix B.1- B.2)*

Theorem 4.3 indicates that the upper bound of the student's expected risk consists of two essential components: (1) the empirical alignment of the first-order expectations ($\hat{R}^s$), and (2) the second-order similarity between the predicted Dirichlet distributions.

#### 4.2.2. Macroscopic First-Order Alignment

Let $\boldsymbol{\alpha}^t$ and $\boldsymbol{\alpha}^s$ denote the concentration parameters produced by the teacher and student branches for different views, respectively. We propose to align both the first-order (centroid) and second-order (distributional) information.

Firstly, we aim to extract the overall characteristics of the second-order distribution to align the expected classification outcomes. For any view, the centroid of the Dirichlet distribution represents the expected categorical probability:

$$
\bar{\boldsymbol{p}} = \mathbb{E}_{\boldsymbol{\mu}\sim\text{Dir}(\boldsymbol{\alpha})}[\boldsymbol{\mu}] = \frac{\boldsymbol{\alpha}}{S},
\tag{10}
$$

The first-order distillation loss can be defined as the KL divergence between the expected probabilities of the teacher and student:

$$
\mathcal{L}_{\text{1st}} = \text{KL}[\bar{\boldsymbol{p}}^t \| \bar{\boldsymbol{p}}^s] = \sum_{k=1}^{K}\frac{\alpha_k^t}{S^t}\log\left(\frac{\alpha_k^t S^s}{S^t \alpha_k^s}\right).
\tag{11}
$$

By optimizing Eq. (11), the student learns to match the teacher's predicted class proportions at a macro level.

#### 4.2.3. Microscopic Second-Order Alignment

Since a fixed centroid does not uniquely determine a Dirichlet distribution, first-order alignment alone is insufficient to capture the fine-grained geometry on the simplex. To achieve precise knowledge transfer of the teacher's uncertainty, we propose a microscopic second-order distillation by directly minimizing the KL divergence between the two Dirichlet distributions:

$$
\mathcal{L}_{\text{2nd}} = \text{KL}\left[\text{Dir}(\boldsymbol{\mu}|\boldsymbol{\alpha}^t) \| \text{Dir}(\boldsymbol{\mu}|\boldsymbol{\alpha}^s)\right].
\tag{12}
$$

The analytical solution for Eq. (12) is derived as:

$$
\begin{aligned}
\mathcal{L}_{\text{2nd}} &= \log\frac{\Gamma(S^t)}{\Gamma(S^s)} - \sum_{k=1}^{K}\log\frac{\Gamma(\alpha_k^t)}{\Gamma(\alpha_k^s)} \\
&\quad + \sum_{k=1}^{K}(\alpha_k^t - \alpha_k^s)\left(\psi(\alpha_k^t) - \psi(S^t)\right),
\end{aligned}
\tag{13}
$$

$\mathcal{L}_{\text{2nd}}$ enforces numerical alignment of the Dirichlet concentration parameters, ensuring that the student faithfully replicates the teacher's predictive nuances and uncertainty patterns across the entire simplex.

*(Derivation.* See the Appendix B.3)

**For the CCA Component**: By combining the macroscopic and microscopic alignment, the total CCA loss is formulated as:

$$
\mathcal{L}_{\text{CCA}} = \mathcal{L}_{\text{1st}} + \mathcal{L}_{\text{2nd}},
\tag{14}
$$

we minimize $\mathcal{L}_{\text{CCA}}$ across different view pairs on the entire dataset $\mathcal{D}_{l\&u}$.

*Table 1.* Performance Comparison of Different Methods on Coarse-grained Datasets

| Methods | CIFAR10 | | | CIFAR100 | | | ImageNet100 | | | All Avg | | |
|---|---|---|---|---|---|---|---|---|---|---|---|---|
| | All | Old | New | All | Old | New | All | Old | New | All | Old | New |
| RankStats+ (Han et al., 2021) | 46.8 | 19.2 | 60.5 | 58.2 | 77.6 | 19.3 | 37.1 | 61.6 | 24.8 | 47.4 | 52.8 | 34.9 |
| UNO+ (Fini et al., 2021) | 68.6 | **98.3** | 53.8 | 69.5 | 80.6 | 47.2 | 70.3 | 95.0 | 57.9 | 69.5 | 91.3 | 53.0 |
| ORCA (Cao et al., 2022) | 81.8 | 86.2 | 79.6 | 69.0 | 77.4 | 52.0 | 73.5 | 92.6 | 63.9 | 74.8 | 85.4 | 65.2 |
| GCD (Vaze et al., 2022) | 91.5 | 97.9 | 88.2 | 73.0 | 76.2 | 66.5 | 74.1 | 89.8 | 66.3 | 79.5 | 88.0 | 73.7 |
| XCon (Fei et al., 2022) | 96.0 | 97.3 | 95.4 | 74.2 | 81.2 | 60.3 | 77.6 | 93.5 | 69.7 | 82.6 | 90.7 | 75.1 |
| DCCL (Pu et al., 2023) | 96.3 | 96.5 | 96.9 | 75.3 | 76.8 | 70.2 | 80.5 | 90.5 | 76.2 | 84.0 | 87.9 | 81.1 |
| GPC (Zhao et al., 2023) | 92.2 | 98.2 | 89.1 | 77.9 | 85.0 | 63.0 | 76.9 | 94.3 | 71.0 | 82.3 | **92.5** | 74.4 |
| InfoSieve (Rastegar et al., 2023) | 94.8 | 97.7 | 93.4 | 78.3 | 82.2 | 70.5 | 80.5 | 93.8 | 73.8 | 84.5 | 91.2 | 79.2 |
| CiPR (Hao et al., 2024) | 97.7 | 97.5 | 97.7 | 81.5 | 82.4 | 79.7 | 80.5 | 84.9 | 78.3 | 86.6 | 88.3 | 85.2 |
| LegoGCD (Cao et al., 2024) | 97.1 | 94.3 | 98.5 | 81.8 | 81.4 | **82.5** | 86.3 | 94.5 | 82.1 | 88.4 | 90.1 | **87.7** |
| CMS (Choi et al., 2024) | - | - | - | 82.3 | 85.7 | 75.5 | 84.7 | **95.6** | 79.2 | - | - | - |
| SPTNet (Wang et al., 2024a) | 97.3 | 95.0 | 98.6 | 81.3 | 84.3 | 75.6 | 85.4 | 93.2 | 81.4 | 88.0 | 90.8 | 85.2 |
| ProtoGCD (Ma et al., 2025) | 97.3 | 95.3 | 98.2 | 81.9 | 82.9 | 80.0 | 84.0 | 92.2 | 79.9 | 87.7 | 90.1 | 86.0 |
| ALLGCD (Cao et al., 2025) | - | - | - | 82.3 | 82.3 | 82.2 | **86.5** | 94.7 | **82.3** | - | - | - |
| SimGCD (Wen et al., 2023) | 97.0 | 93.8 | 98.5 | 79.4 | 78.4 | 81.6 | 83.0 | 93.1 | 77.9 | 86.5 | 88.4 | 86.0 |
| SimGCD+RCA(Ours) | 97.1 | 94.4 | 98.5 | 82.9 | 83.2 | 82.2 | 84.6 | 94.4 | 79.7 | 88.2 | 90.7 | 86.8 |
| *Improvement* | 0.1 | 0.6 | - | 3.5 | 4.8 | 0.6 | 1.6 | 1.3 | 1.8 | 1.7 | 2.3 | 0.8 |
| AFGCD (Xu et al., 2025) | **97.8** | 95.5 | **99.0** | 83.0 | 84.4 | 80.4 | 85.0 | 94.6 | 80.2 | 88.6 | 91.5 | 86.5 |
| AFGCD+RCA(Ours) | **97.8** | 95.5 | **99.0** | **84.8** | **85.9** | **82.5** | 85.9 | 95.2 | 81.1 | **89.5** | 92.2 | 87.5 |
| *Improvement* | - | - | - | 1.8 | 1.5 | 2.1 | 0.9 | 0.6 | 0.9 | 0.9 | 0.7 | 1.0 |

## 4.3. Overall Learning Objective

$\mathcal{L}_{\text{RAC}}$ anchors the evidential network to the ground truth of known classes, preventing belief degeneration, while $\mathcal{L}_{\text{CCA}}$ propagates this reliable uncertainty estimation to the unlabeled data via rigorous distributional alignment. The overall training objective is formulated as follows:

$$\mathcal{L}_{\text{RCA}} = \lambda_1 \mathcal{L}_{\text{RAC}} + \lambda_2 \mathcal{L}_{\text{CCA}}, \tag{15}$$

where $\lambda$ is a trade-off coefficient controlling the strength of different component.

## 5. Experiments

### 5.1. Experiment Setting

**Datasets.** We conduct experiments on commonly used benchmarks, including coarse-grained datasets: CIFAR10 (Krizhevsky et al., 2009), CIFAR100 (Krizhevsky et al., 2009) and ImageNet100 (Deng et al., 2009), as well as fine-grained datasets in the Semantic Shift Benchmark (Vaze et al., 2021): CUB (Wah et al., 2011), Stanford Cars (Krause et al., 2013), and FGVC-Aircraft (Maji et al., 2013).

**Evaluation metrics.** Since ground-truth labels of novel categories are unavailable during training, following standard practice, we assess the GCD performance by performing the Hungarian algorithm (Kuhn, 1955) only once on the entire unlabeled dataset $\mathcal{D}_u$ to compute the clustering accuracy (ACC), denoted as "All". We further report the "Old" and

"New," which represent the performance of known classes and novel classes, respectively. Formally, given the ground-truth labels $y_i$ and the predicted labels $\hat{y}_i$, the clustering accuracy is defined as $\text{ACC} = \frac{1}{|\mathcal{D}_u|} \sum_{i=1}^{|\mathcal{D}_u|} \mathbb{1}\big(y_i = h(\hat{y}_i)\big)$.

**Implementation Details.** Experiments follow the common GCD setup to ensure a fair comparison. We adopt a DINO-pretrained ViT-B/16 backbone, fine-tuning only its final transformer block and using the [CLS] token as the 768-dimensional feature representation. Models are trained with a batch size of 128 for 200 epochs, using a cosine annealing schedule with an initial learning rate of 0.1. The projection space dimension is set to 65,536. As for weak and strong baselines, we follow the original implementations.

### 5.2. Results on Coarse-grained Datasets

In this section, we evaluate the proposed RCA framework on two distinct baselines: the weak baseline SimGCD and the strong baseline AFGCD, which enhances discriminative performance by pruning non-informative regions via an Attention Focusing (AF) mechanism. Our results demonstrate that RCA possesses a robust capability for consistent performance boosting. For instance, on the CIFAR100 dataset, RCA improves the overall accuracy of SimGCD by 3.5%, with significant gains observed in both Old classes (from 78.4% to 83.2%) and New classes (from 81.6% to 82.2%). Moreover, despite the high competitive performance already achieved by AFGCD, our method provides

*Table 2.* Performance Comparison of Different Methods on Fine-grained Datasets

| Methods | CUB | | | Stanford Cars | | | FGVC-Aircraft | | | All Avg | | |
|---|---|---|---|---|---|---|---|---|---|---|---|---|
| | All | Old | New | All | Old | New | All | Old | New | All | Old | New |
| RankStats+ (Han et al., 2021) | 33.3 | 51.6 | 24.2 | 28.3 | 61.8 | 12.1 | 26.9 | 36.4 | 22.2 | 29.5 | 49.9 | 19.5 |
| UNO+ (Fini et al., 2021) | 35.1 | 49.0 | 28.1 | 35.5 | 70.5 | 18.6 | 40.3 | 56.4 | 32.2 | 37.0 | 58.6 | 26.3 |
| ORCA (Cao et al., 2022) | 35.3 | 45.6 | 30.2 | 31.9 | 42.2 | 26.9 | 31.6 | 32.0 | 31.4 | 32.9 | 39.9 | 29.5 |
| GCD (Vaze et al., 2022) | 51.3 | 56.6 | 48.7 | 39.0 | 57.6 | 29.9 | 45.0 | 41.1 | 46.9 | 45.1 | 51.8 | 41.8 |
| XCon (Fei et al., 2022) | 52.1 | 54.3 | 51.0 | 40.5 | 58.8 | 31.7 | 47.7 | 44.4 | 49.4 | 46.8 | 52.5 | 44.0 |
| DCCL (Pu et al., 2023) | 63.5 | 60.8 | 64.9 | 43.1 | 55.7 | 36.2 | - | - | - | - | - | - |
| GPC (Zhao et al., 2023) | 55.4 | 58.2 | 53.1 | 42.8 | 59.2 | 32.8 | 46.3 | 42.5 | 47.9 | 48.2 | 53.3 | 44.6 |
| InfoSieve (Rastegar et al., 2023) | 69.4 | **77.9** | 65.2 | 55.7 | 74.8 | 46.4 | 56.3 | 63.7 | 52.5 | 60.5 | 72.1 | 54.7 |
| CiPR (Hao et al., 2024) | 57.1 | 58.7 | 55.6 | 47.0 | 61.5 | 40.1 | - | - | - | - | - | - |
| LegoGCD (Cao et al., 2024) | 63.8 | 71.9 | 59.8 | 57.3 | 75.7 | 48.4 | 55.0 | 61.5 | 51.7 | 58.7 | 69.7 | 53.3 |
| CMS (Choi et al., 2024) | 68.2 | 76.5 | 64.0 | 56.9 | 76.1 | 47.6 | 56.0 | 63.4 | 52.3 | 60.4 | 72.0 | 54.6 |
| SPTNet (Wang et al., 2024a) | 65.8 | 68.8 | 65.1 | 59.0 | 79.2 | 49.3 | 59.3 | 61.8 | **58.1** | 61.4 | 69.9 | 57.5 |
| ProtoGCD (Ma et al., 2025) | 63.2 | 68.5 | 60.5 | 53.8 | 73.7 | 44.2 | 56.8 | 62.5 | 53.9 | 57.9 | 68.2 | 52.9 |
| ALLGCD (Cao et al., 2025) | 68.4 | 75.1 | 65.1 | 60.5 | 77.0 | 52.5 | 57.4 | 62.4 | 54.9 | 62.1 | 71.5 | 57.5 |
| SimGCD (Wen et al., 2023) | 63.7 | 67.4 | 61.9 | 54.6 | 72.8 | 45.7 | 55.1 | 60.2 | 52.6 | 57.8 | 66.8 | 53.4 |
| SimGCD+RCA(Ours) | 65.2 | 72.4 | 61.7 | 55.6 | 75.6 | 45.9 | 56.8 | 63.0 | 53.7 | 59.2 | 70.3 | 53.8 |
| *Improvement* | 1.5 | 5.0 | -0.2 | 1.0 | 2.8 | 0.2 | 1.7 | 2.8 | 1.1 | 1.4 | 3.5 | 0.4 |
| AFGCD (Xu et al., 2025) | 69.7 | 71.3 | **68.9** | 64.9 | **79.3** | 58.0 | 59.0 | 64.2 | 56.4 | 64.5 | 71.6 | **61.1** |
| AFGCD+RCA(Ours) | **71.2** | 76.4 | 68.6 | **65.0** | 79.2 | **58.2** | **60.1** | **67.1** | 56.6 | **65.4** | **74.2** | **61.1** |
| *Improvement* | 1.5 | 5.1 | -0.3 | 0.1 | -0.1 | 0.2 | 1.1 | 2.9 | 0.2 | 0.9 | 2.6 | - |

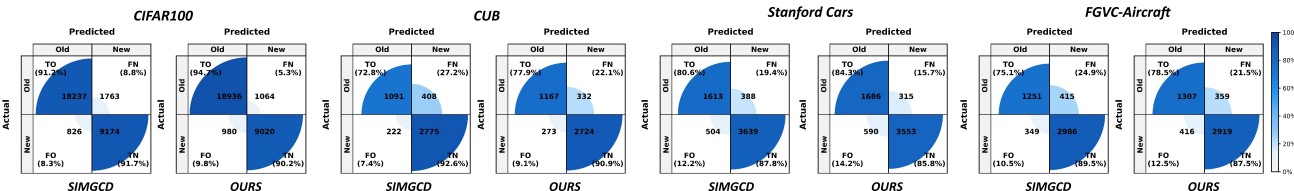

*Figure 4.* Confusion Matrix of GCD on Different Datasets.

further increments on CIFAR100 and ImageNet100. The marginal improvement observed on CIFAR10 is attributed to the inherent simplicity of the dataset; even with entropy regularization, the model maintains high intrinsic certainty for both categories, and the confidence degradation in Old classes remains insignificant, thereby limiting the potential for correctional impact through our RCA framework.

### 5.3. Results on Fine-grained Datasets

Fine-grained classification requires the capture of subtle discriminative features, which presents a significant challenge under the GCD setting and often leads to severe confidence bias. Consequently, RCA yields substantial improvements across different baselines in these challenging scenarios. Experimental results demonstrate that RCA boosts the old-class performance of both SimGCD and AFGCD by approximately 5% on CUB and nearly 3% on FGVC-Aircraft, with negligible sacrifice in new-class accuracy. With the integration of RCA, AFGCD emerges as the most balanced method regarding performance across both old and new classes,

underscoring the superiority of our confidence alignment mechanism in handling high-uncertainty fine-grained tasks.

We further analyze the underlying factors contributing to the performance gains brought by RCA. Figure 4 presents the

*Table 3.* Ablation Results of the Components on Weak Baseline.

| RAC | CCA | CUB | | | CIFAR100 | | |
|---|---|---|---|---|---|---|---|
| | | All | Old | New | All | Old | New |
| ✗ | ✗ | 63.72 | 67.38 | **61.90** | 79.43 | 78.35 | 81.59 |
| ✓ | ✗ | 64.92 | 71.58 | 61.59 | 82.46 | 82.36 | **82.65** |
| ✓ | ✓ | **65.24** | **72.38** | 61.66 | **82.85** | **83.18** | 82.18 |

*Table 4.* Ablation Results of the Components on Strong Baseline.

| RAC | CCA | CUB | | | CIFAR100 | | |
|---|---|---|---|---|---|---|---|
| | | All | Old | New | All | Old | New |
| ✗ | ✗ | 69.66 | 71.25 | **68.87** | 83.02 | 84.36 | 80.35 |
| ✓ | ✗ | 70.73 | 74.92 | 68.64 | 84.41 | 85.15 | **82.94** |
| ✓ | ✓ | **71.20** | **76.38** | 68.60 | **84.76** | **85.87** | 82.54 |

confusion matrices of the baseline and our method across four datasets. A clear reduction in False New (FN) errors is observed, indicating that confidence alignment effectively prevents unlabeled old-class samples from being misclassified as new classes. Although this improvement is accompanied by a moderate increase in False Old (FO) errors, the overall performance on new classes remains largely unaffected. This suggests that RCA does not simply favor old classes, but instead yields a more well-separated and reliable decision boundary between old and new classes, benefiting both stability and plasticity.

*Table 5.* Effect of Different Evidence Construction Functions.

| $\sigma(\cdot)$ | CUB (All) | Stanford Cars (All) | Aircraft (All) |
|---|---|---|---|
| $\mathrm{Exp}(\mathrm{Tanh}(\cdot))$ | **65.24** | **55.55** | **56.79** |
| $\mathrm{Exp}(\cdot)$ | 65.19 | 55.26 | 56.37 |
| $\mathrm{Softplus}(\cdot)$ | 64.68 | 55.44 | 55.87 |

### 5.4. Ablation Study

**The influence of evidence construction methods.** We investigate the impact of different activation functions $\sigma(\cdot)$ for deriving evidence from predictive probabilities. Specifically, we compare several commonly used evidence mappings, including Exp, Softplus, and $\mathrm{Exp}(\tanh(\cdot))$. As reported in Table 5, the evidence construction based on $\mathrm{Exp}(\tanh(\cdot))$ consistently outperforms the alternatives across datasets and evaluation metrics. This superiority can be attributed to the bounded nature of the $\tanh$ function. By constraining the magnitude of the evidence before the exponential transformation, $\tanh(\cdot)$ effectively prevents excessive evidence accumulation that may arise from unbounded mappings.

**The effectiveness of RAC and CCA.** Tables 3 and 4 present systematic ablation studies designed to evaluate the individual and combined effects of the proposed components. Starting from the baseline without any additional modules, incorporating RAC consistently yields stable and significant performance gains across different datasets and baseline settings. On the CUB dataset, RAC leads to a pronounced improvement in old-class accuracy, with only a marginal impact on new-class performance; on CIFAR100, RAC improves both old- and new-class recognition.

Further introducing CCA on top of RAC results in additional, albeit more moderate, performance gains. Specifically, CCA further enhances the old-class performance of both SimGCD and AFGCD. Although a slight degradation in new-class accuracy is observed on CIFAR100, the overall (All) accuracy is consistently improved under both baselines. These results indicate that CCA primarily serves as a complementary regularization mechanism, refining confidence alignment among old-class samples, which is well aligned with our design motivation.

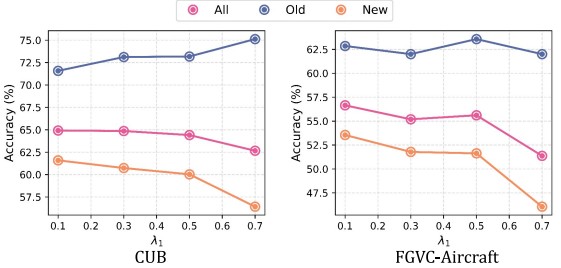

*Figure 5.* Impact of RAC weight coefficient on different datasets.

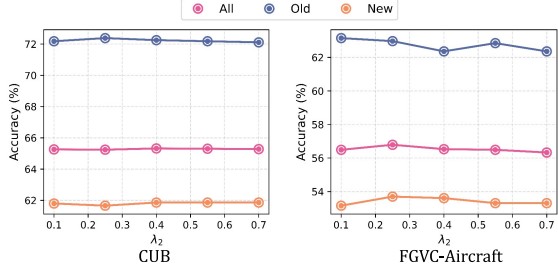

*Figure 6.* Impact of CCA weight coefficient on different datasets.

### 5.5. Hyperparameter Sensitivity

In this section, we investigate the sensitivity of our framework to two core hyperparameters: the anchoring weight $\lambda_1$ for the RAC and the distillation coefficient $\lambda_2$ for the CCA. As shown in Fig 5, increasing $\lambda_1$ from $0.1$ to $0.7$ leads to stable old-class performance on FGVC-Aircraft and a notable improvement on CUB, while consistently degrading new-class accuracy on both datasets. This trend indicates that RAC effectively anchors known semantic evidence, whereas an overly large anchoring weight over-regularizes the representation space and suppresses the learning of new classes. Consequently, $\lambda_1 = 0.1$ provides a favorable trade-off.

By comparison, the CCA coefficient $\lambda_2$ demonstrates strong robustness (Fig 6). Overall performance remains stable over a broad range from $0.1$ to $0.7$. On CUB, the "All" accuracy varies by only $0.1\%$, staying around $65.3\%$. On FGVC-Aircraft, the best performance of $56.79\%$ is achieved at $\lambda_2 = 0.25$. For simplicity, we fix $\lambda_2 = 0.25$.

## 6. Conclusion

In this work, we identify confidence bias induced by entropy regularization as a fundamental cause of the stability–plasticity dilemma in parametric GCD methods. To address this issue, we propose Reliable Confidence Alignment (RCA), which explicitly models epistemic uncertainty using evidential learning to robustly compensate for biased confidence. Extensive experiments on benchmarks of varying granularities demonstrate that RCA substantially mitigates old-class performance degradation while preserving strong novel-class discovery. Our work thus offers a new perspective on tackling GCD problems.

## Acknowledgements

This work is supported by National Science and Technology Major Project (2023ZD0121101), National University of Defense Technology (ZZCX-ZZGC-01-04), Major Fundamental Research Project of Hunan Province (2025JC0005) and National Natural Science Foundation of China (U25A20437).

## Impact Statement

This study fundamentally re-examines the inherent tension between stability and plasticity in Generalized Category Discovery (GCD) through the lens of epistemic uncertainty. By introducing the Reliable Confidence Alignment (RCA) framework, we demonstrate that confidence calibration is not merely an auxiliary post-processing step but can be fundamentally integrated into the model's objective to facilitate reliable open-world learning.

The theoretical insights and empirical evidence provided in this work transcend the specific task of GCD, offering a novel perspective on how learning systems can maintain structural stability within evolving category spaces. Specifically, our analysis of Confidence Bias serves as a diagnostic tool for understanding the side effects of entropy-based regularization, which is a ubiquitous yet double-edged component in modern semi-supervised and self-supervised paradigms. Beyond algorithmic performance, the "plug-and-play" nature of RCA provides a scalable pathway toward constructing trustworthy AI systems, enabling them to autonomously distinguish between the "known-unknowns" and the "unknown-unknowns" with high fidelity.

Ultimately, we believe that uncertainty-aware alignment will inspire more robust research in related domains—including Open-Set Recognition (Geng et al., 2020), Continual Learning (Wang et al., 2024b; Gao et al., 2024; 2025a;b), and Safety-Critical Autonomous Systems—where the predictable reliability of models in complex environments remains a paramount technical challenge.

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

# A. Proof and Derivation of RAC

## A.1. Generative Process and Variational Approximation

We consider a generative process where the latent variable $\boldsymbol{\mu} \in \Delta^{K-1}$ (class probabilities) generates the observed label $\mathbf{y}$. Regarding the RCA component, we have $K = K_{\text{old}}$. The joint probability distribution is factorized as:

$$p(\mathbf{y}, \boldsymbol{\mu}|\mathbf{x}) = p(\mathbf{y}|\boldsymbol{\mu})p(\boldsymbol{\mu}|\mathbf{x}), \qquad (16)$$

where:

- $p(\boldsymbol{\mu}|\mathbf{x}) = \text{Dir}(\boldsymbol{\mu}|\mathbf{1})$ is the uniform Dirichlet prior, representing maximum entropy before evidence acquisition.

- $p(\mathbf{y}|\boldsymbol{\mu}) = \text{Mult}(\mathbf{y}|\boldsymbol{\mu})$ is the multinomial likelihood.

To approximate the intractable true posterior $p(\boldsymbol{\mu}|\mathbf{x}, \mathbf{y})$, we introduce a variational distribution $q_\theta(\boldsymbol{\mu}|\mathbf{x})$, which is parameterized by the outputs of model: $\boldsymbol{\alpha} = f_\theta(\mathbf{x}) + \mathbf{1}$ as a Dirichlet distribution:

$$q_\theta(\boldsymbol{\mu}|\mathbf{x}) = \text{Dir}(\boldsymbol{\mu}|\boldsymbol{\alpha}). \qquad (17)$$

The subsequent derivation is based on the content of this section.

## A.2. Proof of the Evidence Lower Bound

Our goal is to maximize the log-marginal likelihood $\log p(\mathbf{y}|\mathbf{x})$ for evidence acquisition. By introducing the variational posterior, we can reformulate the evidence using **Jensen's Inequality**:

$$\begin{aligned}
\log p(\mathbf{y}|\mathbf{x}) &= \log \int_{\Delta^{K-1}} p(\mathbf{y}, \boldsymbol{\mu}|\mathbf{x})d\boldsymbol{\mu} \\
&= \log \int_{\Delta^{K-1}} q_\theta(\boldsymbol{\mu}|\mathbf{x}) \frac{p(\mathbf{y}, \boldsymbol{\mu}|\mathbf{x})}{q_\theta(\boldsymbol{\mu}|\mathbf{x})}d\boldsymbol{\mu} \\
&\geq \int_{\Delta^{K-1}} q_\theta(\boldsymbol{\mu}|\mathbf{x}) \log \frac{p(\mathbf{y}, \boldsymbol{\mu}|\mathbf{x})}{q_\theta(\boldsymbol{\mu}|\mathbf{x})}d\boldsymbol{\mu} \quad . \quad (18)
\end{aligned}$$

The right-hand side is the Evidence Lower Bound (ELBO), denoted as $\mathcal{L}_{\text{ELBO}}$. We can further decompose the gap between the true evidence and the ELBO:

$$\begin{aligned}
\log p(\mathbf{y}|\mathbf{x}) - \mathcal{L}_{\text{ELBO}} &= \mathbb{E}_{q_\theta}\left[\log p(\mathbf{y}|\mathbf{x}) - \log \frac{p(\mathbf{y}, \boldsymbol{\mu}|\mathbf{x})}{q_\theta(\boldsymbol{\mu}|\mathbf{x})}\right] \\
&= \mathbb{E}_{q_\theta}\left[\log \frac{p(\mathbf{y}|\mathbf{x})q_\theta(\boldsymbol{\mu}|\mathbf{x})}{p(\mathbf{y}, \boldsymbol{\mu}|\mathbf{x})}\right] \\
&= \mathbb{E}_{q_\theta}\left[\log \frac{q_\theta(\boldsymbol{\mu}|\mathbf{x})}{p(\boldsymbol{\mu}|\mathbf{x}, \mathbf{y})}\right] \\
&= \text{KL}\left[q_\theta(\boldsymbol{\mu}|\mathbf{x})\|p(\boldsymbol{\mu}|\mathbf{x}, \mathbf{y})\right]. \qquad (19)
\end{aligned}$$

This confirms Eq. (3) in the main text.

## A.3. Proof of Theorem 4.2

According to A.1 and A.2, to maximize the evidence, we can maximize the ELBO. Expanding $\mathcal{L}_{\text{ELBO}}$, we have:

$$\begin{aligned}
\mathcal{L}_{\text{ELBO}} &= \int_{\Delta^{K-1}} q_\theta(\boldsymbol{\mu}|\mathbf{x}) \log \frac{p(\mathbf{y}|\boldsymbol{\mu})p(\boldsymbol{\mu}|\mathbf{x})}{q_\theta(\boldsymbol{\mu}|\mathbf{x})}d\boldsymbol{\mu} \\
&= \underbrace{\mathbb{E}_{q_\theta}[\log p(\mathbf{y}|\boldsymbol{\mu})]}_{\text{Expected Log-Likelihood}} - \underbrace{\mathbb{E}_{q_\theta}\left[\log \frac{q_\theta(\boldsymbol{\mu}|\mathbf{x})}{p(\boldsymbol{\mu}|\mathbf{x})}\right]}_{\text{KL Divergence}}. \quad (20)
\end{aligned}$$

Minimizing the negative ELBO establishes the theoretical foundation for our objective:

$$\mathcal{L}(\theta) = \mathbb{E}q\theta[-\log p(\mathbf{y}|\boldsymbol{\mu})] + \text{KL}\left[q_\theta(\boldsymbol{\mu}|\mathbf{x})|p(\boldsymbol{\mu}|\mathbf{x})\right]. \quad (21)$$

By instantiating the distributions with the Dirichlet forms defined in Eq. (16) and Eq. (17), we arrive at the final learning objective:

$$\mathcal{L}(\theta) = \mathbb{E}_{\boldsymbol{\mu}\sim\text{Dir}(\boldsymbol{\alpha})}[-\log p(\mathbf{y}|\boldsymbol{\mu})] + \text{KL}\left[\text{Dir}(\boldsymbol{\mu}|\boldsymbol{\alpha})|\text{Dir}(\boldsymbol{\mu}|\mathbf{1})\right]. \quad (22)$$

This confirms Eq. (5) in the main text.

## A.4. Derivation of The Expected Risk.

The detailed derivation of Eq. (6) in the main text is as follows:

$$\begin{aligned}
\mathcal{L}_{\text{risk}}(\boldsymbol{\alpha}_i) &= \mathbb{E}_{\text{Dir}(\boldsymbol{\mu}|\boldsymbol{\alpha}_i)}[-\log p(\mathbf{y}_i \mid \boldsymbol{\mu})] \\
&= \mathbb{E}_{\text{Dir}(\boldsymbol{\mu}|\boldsymbol{\alpha}_i)}\left[-\log \prod_{k=1}^{K} \mu_k^{y_{ik}}\right] \\
&= \int_{\Delta^{K-1}} \left[-\sum_{k=1}^{K} y_{ik} \log(\mu_k)\right] \frac{1}{B(\boldsymbol{\alpha}_i)} \prod_{k=1}^{K} \mu_k^{\alpha_{ik}-1} d\boldsymbol{\mu} \\
&= \sum_{k=1}^{K} y_{ik}\left[-\int_{\Delta^{K-1}} \log(\mu_k) \frac{1}{B(\boldsymbol{\alpha}_i)} \prod_{j=1}^{K} \mu_j^{\alpha_{ij}-1} d\boldsymbol{\mu}\right] \\
&= \sum_{k=1}^{K} y_{ik}\left(\psi(S_i) - \psi(\alpha_{ik})\right).
\end{aligned} \quad (23)$$

# B. Proof and Derivation of CCA

## B.1. Risk Definitions

In our Reliable Confidence Alignment (RCA) framework, the Dirichlet distributions predicted by the teacher and student networks, denoted as $q^t = \text{Dir}(\boldsymbol{\mu} \mid \boldsymbol{\alpha}_i^t)$ and $q^s = \text{Dir}(\boldsymbol{\mu} \mid \boldsymbol{\alpha}_i^s)$, represent posterior beliefs over the probability simplex $\Delta^{K-1}$.

For a single instance $\mathbf{x}_i$, the student's misclassification risk relative to the teacher is defined as the expected cross-entropy. Let $\bar{\boldsymbol{p}}_i^t = \mathbb{E}_{q^t}[\boldsymbol{\mu}] = \boldsymbol{\alpha}_i^t/S_i^t$ and $\bar{\boldsymbol{p}}_i^s = \mathbb{E}_{q^s}[\boldsymbol{\mu}] = \boldsymbol{\alpha}_i^s/S_i^s$ be the first-order moments (centroids) of the respec-

tive distributions. The risk for sample $i$ is:

$$
\begin{aligned}
R_i^s &= -\sum_{k=1}^{K} \bar{p}_{ik}^t \log \bar{p}_{ik}^s \\
&= \mathbb{E}_{\boldsymbol{\mu} \sim q^t}\left[-\sum_{k=1}^{K} \mu_k \log \mathbb{E}_{\boldsymbol{\mu}' \sim q^s}[\mu_k']\right] \\
&= \mathbb{E}_{\boldsymbol{\mu} \sim q^t}[R_i^s(\boldsymbol{\mu})],
\end{aligned}
\tag{24}
$$

where $R_i^s(\boldsymbol{\mu}) = -\sum_{k=1}^{K} \mu_k \log \bar{p}_{ik}^s$ represents the loss for a specific probability vector $\boldsymbol{\mu}$ drawn from the teacher's posterior.

Given a training set $\mathcal{D}_{l\&u}$ of size $N$, the **empirical risk** $\hat{R}^s$ and the **expected risk** $R^s$ over the underlying data distribution $\mathcal{D}$ are:

$$
\hat{R}^s = \frac{1}{N}\sum_{i=1}^{N}\mathbb{E}_{\boldsymbol{\mu} \sim q^t}[R_i^s(\boldsymbol{\mu})], \quad R^s = \mathbb{E}_{\mathbf{x} \sim \mathcal{D}}[R_i^s]. \tag{25}
$$

## B.2. Proof of Theorem 4.3

To bound the expected risk $R^s$, we leverage the PAC-Bayesian framework. Consider the following non-negative random variable dependent on the training set $\mathcal{D}_{l\&u}$:

$$
X(\mathcal{D}_{l\&u}) = \sum_{i=1}^{N}\mathbb{E}_{\boldsymbol{\mu} \sim q^s}\left[e^{\frac{N}{\gamma}(R^s - R_i^s(\boldsymbol{\mu}))}\right]. \tag{26}
$$

By invoking **Markov's Inequality**, for any $\delta \in (0, 1]$, the following holds with probability at least $1 - \delta$:

$$
\begin{aligned}
&\sum_{i=1}^{N}\mathbb{E}_{\boldsymbol{\mu} \sim q^s}\left[e^{\frac{N}{\gamma}(R^s - R_i^s(\boldsymbol{\mu}))}\right] \\
&\leq \frac{1}{\delta}\mathbb{E}_{\mathcal{D}_{l\&u} \sim \mathcal{D}^N}\left[\sum_{i=1}^{N}\mathbb{E}_{\boldsymbol{\mu} \sim q^s}\left[e^{\frac{N}{\gamma}(R^s - R_i^s(\boldsymbol{\mu}))}\right]\right].
\end{aligned}
\tag{27}
$$

Let the right-hand side of Eq. (27) be denoted as $1/\delta \cdot \text{Const} = \mathcal{C}'(\delta)$.

To introduce the divergence between the student and teacher distributions, we apply a **change of measure** (importance sampling) on the left-hand side:

$$
\mathbb{E}_{\boldsymbol{\mu} \sim q^s}\left[e^{\frac{N}{\gamma}(R^s - R_i^s(\boldsymbol{\mu}))}\right] = \mathbb{E}_{\boldsymbol{\mu} \sim q^t}\left[\frac{q^s(\boldsymbol{\mu})}{q^t(\boldsymbol{\mu})}e^{\frac{N}{\gamma}(R^s - R_i^s(\boldsymbol{\mu}))}\right]. \tag{28}
$$

Substituting this back into the inequality and taking the logarithm on both sides:

$$
\log\left(\sum_{i=1}^{N}\mathbb{E}_{\boldsymbol{\mu} \sim q^t}\left[\frac{q^s(\boldsymbol{\mu})}{q^t(\boldsymbol{\mu})}e^{\frac{N}{\gamma}(R^s - R_i^s(\boldsymbol{\mu}))}\right]\right) \leq \log(\mathcal{C}'(\delta)). \tag{29}
$$

By applying **Jensen's Inequality** to the concave logarithm function, we have:

$$
\sum_{i=1}^{N}\mathbb{E}_{\boldsymbol{\mu} \sim q^t}\left[\log\frac{q^s(\boldsymbol{\mu})}{q^t(\boldsymbol{\mu})} + \frac{N}{\gamma}(R^s - R_i^s(\boldsymbol{\mu}))\right] \leq \log(\mathcal{C}'(\delta)). \tag{30}
$$

Recalling the definition of Kullback-Leibler divergence, $\mathbb{E}_{q^t}[\log\frac{q^t}{q^s}] = \text{KL}(q^t\|q^s)$, the inequality simplifies to:

$$
-\sum_{i=1}^{N}\text{KL}[q^t\|q^s] + \frac{N^2}{\gamma}(R^s - \hat{R}^s) \leq \log(\mathcal{C}'(\delta)). \tag{31}
$$

Rearranging the terms to isolate the expected risk $R^s$:

$$
R^s \leq \hat{R}^s + \frac{\gamma}{N}\sum_{i=1}^{N}\text{KL}[\text{Dir}(\boldsymbol{\alpha}_i^t)\|\text{Dir}(\boldsymbol{\alpha}_i^s)] + C(\delta), \tag{32}
$$

where $C(\delta) = \frac{\gamma}{N^2}\log(\mathcal{C}'(\delta))$ is a constant term. This completes the proof.

## B.3. Derivation of Second-Order Alignment

The detailed derivation of Eq. (12) in the main text is as follows:

$$
\begin{aligned}
\mathcal{L}_{\text{2nd}} &= \text{KL}\left[\text{Dir}(\boldsymbol{\mu}|\boldsymbol{\alpha}^t) \,\|\, \text{Dir}(\boldsymbol{\mu}|\boldsymbol{\alpha}^s)\right] \\
&= \int \text{Dir}(\boldsymbol{\mu} \mid \boldsymbol{\alpha}^t)\log\left(\frac{\text{Dir}(\boldsymbol{\mu} \mid \boldsymbol{\alpha}^t)}{\text{Dir}(\boldsymbol{\mu} \mid \boldsymbol{\alpha}^s)}\right)\mathrm{d}\boldsymbol{\mu} \\
&= \int \frac{1}{B(\boldsymbol{\alpha}^t)}\prod_{k=1}^{K}\mu_k^{\alpha_k^t - 1}\left[\log\frac{B(\boldsymbol{\alpha}^s)}{B(\boldsymbol{\alpha}^t)}\right. \\
&\quad \left.+ \sum_{k=1}^{K}(\alpha_k^t - \alpha_k^s)\log\mu_k\right]\mathrm{d}\boldsymbol{\mu} \\
&= \log\frac{B(\boldsymbol{\alpha}^s)}{B(\boldsymbol{\alpha}^t)} + \sum_{k=1}^{K}(\alpha_k^t - \alpha_k^s)\,\mathbb{E}_{\boldsymbol{\alpha}^t}[\log\mu_k] \\
&= \left(\log\Gamma(S^t) - \sum_{k=1}^{K}\log\Gamma(\alpha_k^t)\right) \\
&\quad - \left(\log\Gamma(S^s) - \sum_{k=1}^{K}\log\Gamma(\alpha_k^s)\right) \\
&\quad + \sum_{k=1}^{K}(\alpha_k^t - \alpha_k^s)\bigl(\psi(\alpha_k^t) - \psi(S^t)\bigr) \\
&= \log\frac{\Gamma(S^t)}{\Gamma(S^s)} - \sum_{k=1}^{K}\log\frac{\Gamma(\alpha_k^t)}{\Gamma(\alpha_k^s)} \\
&\quad + \sum_{k=1}^{K}(\alpha_k^t - \alpha_k^s)\bigl(\psi(\alpha_k^t) - \psi(S^t)\bigr).
\end{aligned}
\tag{33}
$$

# C. Generic Components of Generalized Category Discovery (SimGCD)

This section summarizes a representative parametric framework for Generalized Category Discovery (GCD), namely SimGCD (Wen et al., 2023), which serves as the primary reference model in our empirical analysis. Such methods are typically composed of two components: a representation learning module and a classifier learning module.

## C.1. Representation Learning

The goal of representation learning is to construct a discriminative feature space from a mixture of labeled and unlabeled data. SimGCD achieves this by jointly leveraging self-supervised and supervised contrastive objectives, thereby encouraging the class separability.

### C.1.1. SELF-SUPERVISED CONTRASTIVE LEARNING

Given a mini-batch $\mathcal{B}$, each image is transformed into two views $\mathbf{x}_i$ and $\mathbf{x}'_i$ via data augmentation. The views are encoded by a backbone network $f_\theta(\cdot)$ followed by a projection head $g(\cdot)$, yielding $\ell_2$-normalized embeddings $\mathbf{z}_i = g(f_\theta(\mathbf{x}))$. The self-supervised contrastive loss is formulated as

$$\mathcal{L}_{\text{unsup}} = -\sum_{i \in \mathcal{B}} \log \frac{\exp(\mathbf{z}_i^\top \mathbf{z}'_i / \tau_u)}{\sum_{j \neq i} \exp(\mathbf{z}_i^\top \mathbf{z}'_j / \tau_u)}, \quad (34)$$

where $\tau_u$ denotes a temperature parameter.

### C.1.2. SUPERVISED CONTRASTIVE LEARNING

For labeled samples, category information is exploited to further enhance feature discrimination. Let $\mathcal{N}_i$ denote the set of indices in $\mathcal{B}_l$ that share the same class label as sample $i$. The supervised contrastive objective is defined as

$$\mathcal{L}_{\text{sup}} = -\sum_{i \in \mathcal{B}_l} \frac{1}{|\mathcal{N}_i|} \sum_{q \in \mathcal{N}_i} \log \frac{\exp(\mathbf{z}_i^\top \mathbf{z}_q / \tau_s)}{\sum_{j \neq i} \exp(\mathbf{z}_i^\top \mathbf{z}'_j / \tau_s)}, \quad (35)$$

### C.1.3. REPRESENTATION LEARNING OBJECTIVE

The two contrastive losses are combined through a weighted sum, yielding the overall representation learning objective:

$$\mathcal{L}_{\text{rep}} = (1 - \lambda_{\text{sim}})\mathcal{L}_{\text{unsup}} + \lambda_{\text{sim}}\mathcal{L}_{\text{sup}}. \quad (36)$$

where $\lambda_{\text{sim}}$ is a hyperparameter controlling the weight of the supervised loss.

## C.2. Classifier Learning

SimGCD adopts a parametric prototype-based classifier trained with a hybrid supervised and self-supervised optimization strategy, where the total number of categories $K$ is assumed to be known a priori.

### C.2.1. PROTOTYPE-BASED CLASSIFICATION AND SELF-DISTILLATION

A learnable prototype vector $\mathbf{c}_k$ is maintained for each category, and all prototypes are collected into $\mathbf{C} = [\mathbf{c}_1, \ldots, \mathbf{c}_K]$. Given a feature representation $\mathbf{h}_i = f_\theta(\mathbf{x}_i)$, the predictive distribution is given by:

$$\boldsymbol{p}_i^{(k)} = \frac{\exp\left(\frac{1}{\tau_c}(\boldsymbol{h}_i/||\boldsymbol{h}_i||_2)^\top(\boldsymbol{c}_k/||\boldsymbol{c}_k||_2)\right)}{\sum_{k'} \exp\left(\frac{1}{\tau_c}(\boldsymbol{h}_i/||\boldsymbol{h}_i||_2)^\top(\boldsymbol{c}_{k'}/||\boldsymbol{c}_{k'}||_2)\right)}, \quad (37)$$

where $\tau_c$ controls the sharpness of the predicted distribution. By generating an additional augmented view $\mathbf{x}'_i$ with a lower temperature $\tau_t$, we obtain a sharper soft label $\mathbf{q}'_i$, which serves as a supervisory signal for self-distillation.

### C.2.2. CLASSIFICATION LOSSES

For labeled samples, standard cross-entropy loss is employed. For unlabeled samples, the model is trained to match the sharpened target soft label (self-distillation):

$$\mathcal{L}_{\text{cls}}^{\text{lab}} = \frac{1}{|\mathcal{B}_l|} \sum_{i \in \mathcal{B}_l} \mathcal{L}_{\text{ce}}(\mathbf{y}_i, \mathbf{p}_i), \quad (38)$$

$$\mathcal{L}_{\text{cls}}^{\text{unl}} = \frac{1}{|\mathcal{B}|} \sum_{i \in \mathcal{B}} \mathcal{L}_{\text{ce}}(\mathbf{q}'_i, \mathbf{p}_i), \quad (39)$$

where $\mathbf{y}_i$ denotes the one-hot encoded ground-truth label.

### C.2.3. ENTROPY REGULARIZATION

To prevent prediction collapse at the batch level, an entropy regularization term is introduced based on the mean predictive distribution. Specifically, $\bar{\mathbf{p}} = \frac{1}{2|\mathcal{B}|} \sum_{i \in \mathcal{B}}(\mathbf{p}_i + \mathbf{p}'_i)$ represents the mean prediction of a batch, and the entropy is defined as: $H(\bar{\boldsymbol{p}}) = -\sum_k \bar{\boldsymbol{p}}^{(k)} \log \bar{\boldsymbol{p}}^{(k)}$. Since training is performed by minimizing the objective function, we formulate the entropy regularization as the negative entropy loss:

$$\mathcal{L}_{\text{ER}} = -H(\bar{\mathbf{p}}) = \sum_{k=1}^{K} \bar{p}^{(k)} \log \bar{p}^{(k)}. \quad (40)$$

### C.2.4. CLASSIFIER LEARNING OBJECTIVE

The overall objective for classifier learning is given by

$$\mathcal{L}_{\text{cls}} = \lambda_{\text{sim}}\mathcal{L}_{\text{cls}}^{\text{lab}} + (1 - \lambda_{\text{sim}})\mathcal{L}_{\text{cls}}^{\text{unl}} + \mathcal{L}_{\text{ER}}. \quad (41)$$

## C.3. Overall Training Objective

Finally, the complete optimization objective of SimGCD is formulated as

$$\mathcal{L}_{\text{SimGCD}} = \mathcal{L}_{\text{rep}} + \mathcal{L}_{\text{cls}}. \quad (42)$$

## D. AFGCD

AFGCD extends the parametric GCD framework by introducing two lightweight modules that enhance token-level feature selection in vision transformers:

- **Token Importance Measurement (TIME)**: produces importance scores for each patch token, allowing the model to focus on the most informative regions of an image.

- **Token Adaptive Pruning (TAP)**: removes less relevant tokens based on their importance scores, improving computational efficiency and representation quality.

To guide the learning of token importance, TIME module is equipped with an auxiliary classification head, optimized using the standard cross-entropy loss. Specifically, for the $\ell$-th TIME module, given a batch of samples with one-hot labels $\mathbf{y}_i$ and predictions $\mathbf{p}_i^\ell$, the loss is

$$\mathcal{L}_{\text{ce}}^\ell = -\frac{1}{|\mathcal{B}_l|} \sum_{i \in \mathcal{B}_l} \sum_{k=1}^{|\mathcal{Y}_l|} y_{ik} \log p_{ik}. \tag{43}$$

The overall training objective of AFGCD combines the original SimGCD loss with the auxiliary TIME losses:

$$\mathcal{L}_{\text{AFGCD}} = \mathcal{L}_{\text{SimGCD}} + \lambda_{\text{af}} \sum_{\ell=1}^{L-1} \mathcal{L}_{\text{ce}}^\ell, \tag{44}$$

where $L$ is the number of TIME-equipped blocks and $\lambda_{\text{af}}$ is a balancing hyperparameter.

## E. More Implementation Details

### E.1. The Details of Datasets

Table 6 presents the class and sample count splits of the labeled and unlabeled sets for each dataset. To ensure a fair comparison and enable systematic analysis, we adopt the same data partitioning protocol as established in prior GCD literature (Vaze et al., 2022; Wen et al., 2023; Ma et al., 2025; Xu et al., 2025).

### E.2. The Details of Hyper-parameters

Following SimGCD and AFGCD, we set the $\lambda_{\text{sim}} = 0.35$. The temperatures for contrastive learning are $\tau_u = 0.07$ and $\tau_s = 1.0$. For the classification objective, $\tau_c = 0.1$, and the self-distillation temperature $\tau_t$ is initialized at 0.07 and gradually decreased to 0.04 using a cosine schedule during the first 30 training epochs. For $\lambda_{\text{af}}$, we adopt the default value of 0.05 as specified in the original work. All experiments are performed on an NVIDIA GeForce RTX 4090 GPU.

*Table 6.* Dataset splits of GCD setting.

| Dataset | $|\mathcal{D}_l|$ | $|\mathcal{Y}_l|$ | $|\mathcal{D}_u|$ | $|\mathcal{Y}_u|$ |
|---|---|---|---|---|
| CIFAR10 | 12.5 K | 5 | 37.5 K | 10 |
| CIFAR100 | 20.0 K | 80 | 30.0 K | 100 |
| ImageNet100 | 31.9 K | 50 | 95.3 K | 100 |
| CUB | 1.5 K | 100 | 4.5 K | 200 |
| Stanford Cars | 2.0 K | 98 | 6.1 K | 196 |
| FGVC-Aircraft | 1.7 K | 50 | 5.0 K | 100 |

## F. More Experimental Results

To further validate that the observed Confidence Bias is not an isolated issue specific to SimGCD, but rather reflects a ubiquitous limitation inherent in parametric GCD frameworks, we extend our empirical analysis to the more advanced AFGCD baseline. As illustrated in Figure 7, we observe similar misalignment patterns to those discussed in Sec. 3.2. This suggests that although AFGCD generates higher-quality pseudo-labels through its Attention Focusing (AF) mechanism and achieves superior performance, it still fails to mitigate the intrinsic bias stemming from entropy regularization.

By integrating our proposed RCA framework, a significant alignment is achieved between the confidence scores of labeled and unlabeled instances within the known categories. This observation directly accounts for the performance gains observed on AFGCD across both coarse-grained (Table 1) and fine-grained datasets (Table 2). Notably, on the fine-grained CUB dataset, RCA demonstrates a more pronounced alignment effect when applied to the stronger AFGCD baseline compared to SimGCD. This synergy further underscores that our method is highly compatible with, and can effectively enhance, various parametric GCD approaches that prioritize representation and label quality, highlighting the broad potential impact of RCA.

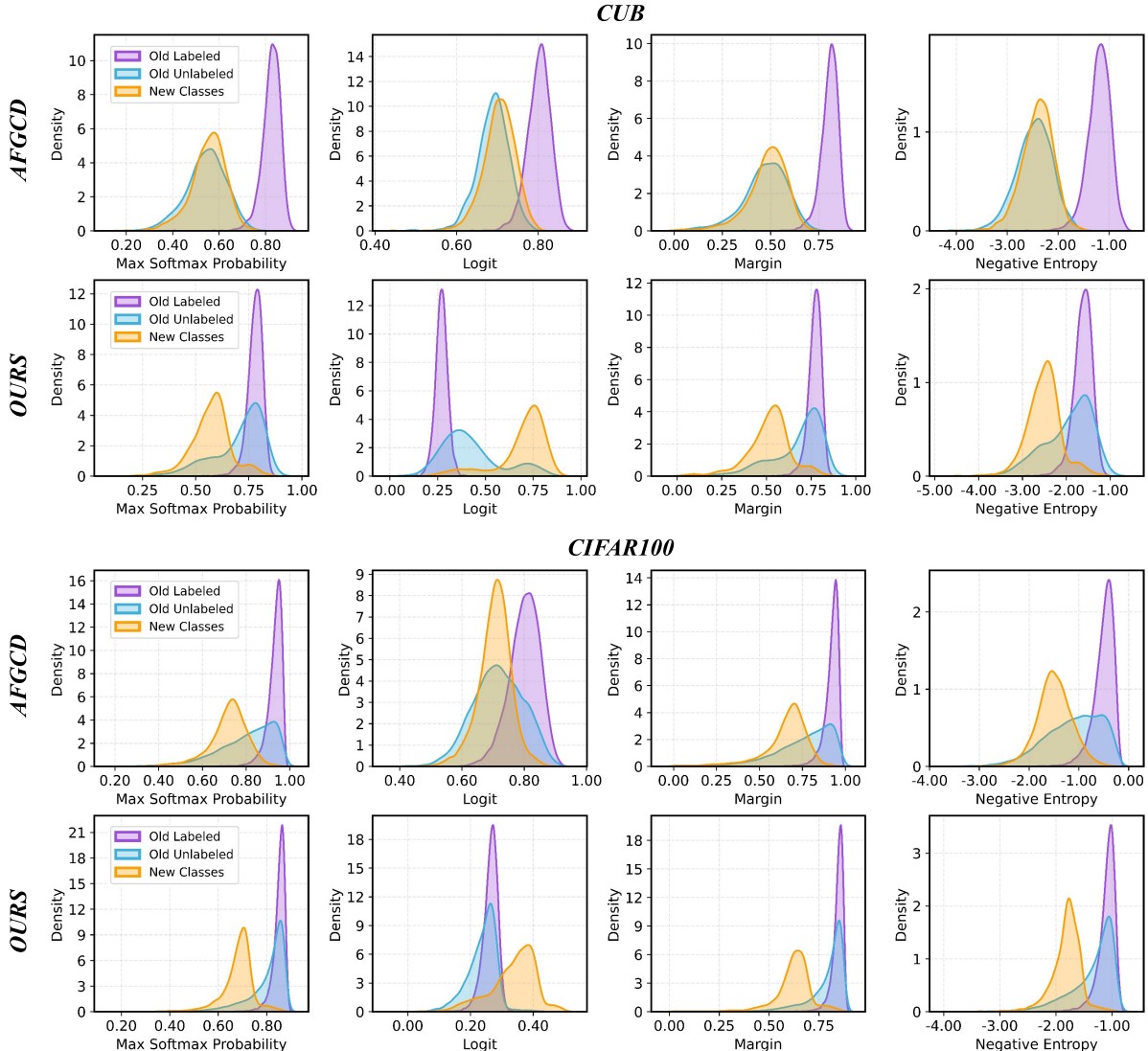

*Figure 7.* Confidence gap of different metrics across multiple datasets, comparing AFGCD and AFGCD+RCA (our method).

