# OpenReview forum: "Reliable Confidence Alignment for Generalized Category Discovery"
_ICML.cc/2026/Conference — ICML 2026 regular_

### Official Review · Reviewer_oqsm · 2026-03-08

**Soundness:** 3
**Presentation:** 3
**Significance:** 2
**Originality:** 2
**Overall Recommendation:** 4
**Confidence:** 3

**Summary:**

This paper proposes an evidential deep learning (EDL)-based framework for generalized category discovery that addresses confidence mismatch between labeled and unlabeled old-class samples. By aligning confidence across views and anchoring reliable certainty on labeled data, the method improves old-class performance without substantially degrading novel-class discovery.

**Compliance With Llm Reviewing Policy:**

Affirmed.

**Final Justification:**

In my original review, I found the paper’s main contributions to be novel and technically sound, but I had some concerns about specific design choices, such as the use of EDL as the UQ model and the validity of second-order alignment.

The authors provided a comprehensive rebuttal that addressed these concerns and convinced me. Accordingly, I have increased my score to 4 (Weak Accept).

**Key Questions For Authors:**

* **1. (Choice of Base UQ Model)** What is the rationale for choosing the original EDL model (Sensoy et al., 2018) as the base uncertainty estimator for RAC? In particular, what advantage does it offer over more recent EDL variants (including those cited in the *Weaknesses* section) or other uncertainty estimation approaches, such as deterministic single-pass UQ methods or Bayesian methods, which could potentially serve a similar role in this framework?


* **2. (Generality as a Plug-in Framework)** Could the authors provide additional experiments combining RCA with other strong GCD baselines, such as ProtoGCD and ALLGCD? Since the proposed method appears to be plug-and-play, it would strengthen the empirical validation to show that RCA consistently improves performance when attached to a broader range of competitive baselines, rather than only in a limited setting.


* **3. (Definition of Different Views and Augmentation Protocol)** What exactly constitutes the *different view* used in Section 4.2? Please clarify what types of data augmentation are applied, whether they follow a standard protocol or a custom setup, and whether any additional samples are required beyond the original training data. Since the effectiveness of the proposed first- and second-order alignment may depend on the augmentation strategy, this design choice should be specified more clearly.

* **4. (On the uncertainty quality)** Since the method is motivated from an uncertainty estimation perspective, could the authors provide a more direct evaluation of uncertainty quality, beyond downstream clustering/classification performance? For instance, calibration metrics or selective prediction behavior may provide stronger evidence that RCA improves confidence estimation, rather than merely leading to higher classification accuracy.

Overall, I believe this paper makes a valuable contribution, although the current presentation still leaves several important issues insufficiently clarified. If the authors can adequately address the concerns above, I would be happy to raise my score.

**Limitations:**

The manuscript does not explicitly discuss limitations or potential negative societal impacts. However, given the scope and nature of the proposed method, there do not appear to be any critical limitations or significant societal risks that require dedicated discussion.

**Strengths And Weaknesses:**

### **Strengths**

* The paper tackles an important and practically relevant challenge in GCD, and the motivation is generally clear and easy to follow.
* The idea of applying an uncertainty-aware framework (EDL) to achieve confidence alignment is sound.

### **Weaknesses**

While the motivation is clear, the specific design choices appear somewhat incremental and ad hoc, which I see as the main limitation of the paper.

**(1. Choice of EDL for RAC is insufficiently justified)**

While adopting EDL [1] is a reasonable and standard choice, the paper does not clearly explain why EDL is particularly well-suited for the proposed confidence alignment objective in GCD. More importantly, EDL has known theoretical and empirical limitations, and there exist both more recent EDL variants (e.g., PostNet [2], I-EDL [3], R-EDL [4], DAEDL [5], Re-EDL [6], F-EDL [7]) and other deterministic uncertainty estimation approaches (e.g., SNGP [8], DUQ [9], DDU [10]) that could plausibly serve a similar role.

Since this is not primarily an uncertainty quantification (UQ) paper, I do not expect an exhaustive empirical comparison against all UQ methods, nor do I argue that the authors must adopt the latest UQ approach. However, the paper should at least discuss relevant alternatives in the Related Work section, review how these methods have been applied in related settings (see, for example, survey papers such as [11]), and more clearly explain why EDL, in particular, is a suitable choice for achieving the proposed confidence alignment.

**(2. RAC mainly adopts existing EDL components rather than introducing a fundamentally new formulation.)**

To my understanding, RAC is better characterized as an adaptation of standard EDL components to the GCD setting, rather than as a newly developed formulation in itself. In particular, its main ingredients—including the variational perspective, expected risk term, and KL-divergence-based regularization—have already been introduced and motivated in prior EDL literature. While I appreciate the effort to incorporate these ideas into GCD, the paper should more clearly distinguish between what is adopted from existing EDL formulations and what is newly contributed in this work.


**(3. The CCA alignment design is reasonable, but somewhat ad hoc.)**

The overall idea of aligning confidence across views is sound, but some aspects of the formulation remain unclear. In particular, it is not explicitly stated what the *different views* refer to in CCA. I assume they correspond to two augmentations of the same input, but this should be stated clearly. In addition, I am not fully convinced that second-order distributional information should always be aligned so strictly across views. Depending on the type and strength of augmentation, it may be overly restrictive to require near-perfect agreement not only in the predictive mean but also in the full uncertainty structure.

Moreover, I also have some concerns about the experiment part.

**(4. Incremental Performance Gain and Limited Comparisons)**

The performance improvements over strong baselines are relatively modest overall, and the method does not consistently outperform competitive alternatives such as ALLGCD on ImageNet100. While I do not believe that state-of-the-art performance on every benchmark is a strict requirement for acceptance, I find the experimental validation somewhat limited.
In particular, since the proposed method is framed as a plug-in module, the paper should evaluate it on a broader set of competitive GCD baselines (e.g., ALLGCD, ProtoGCD, SPTNet). Without such comparisons, it is difficult to assess whether the observed gains reflect a generally useful module or are instead somewhat tied to the specific base frameworks used in the paper.


#### **References**

[1] Sensoy et al., Evidential Deep Learning to Quantify Classification Uncertainty, NeurIPS 2018.

[2] Charpentier et al., Posterior Network: Uncertainty Estimation Without OOD Samples via Density-Based Pseudo-Counts, NeurIPS 2020.

[3] Deng et al., Uncertainty Estimation by Fisher Information-Based Evidential Deep Learning, ICML 2023.

[4] Chen et al., R-EDL: Relaxing Nonessential Settings of Evidential Deep Learning, ICLR 2024.

[5] Yoon and Kim, Uncertainty Estimation by Density-Aware Evidential Deep Learning, ICML 2024.

[6] Chen et al., Revisiting Essential and Nonessential Settings of Evidential Deep Learning, IEEE TPAMI 2025.

[7] Yoon and Kim, Uncertainty Estimation by Flexible Evidential Deep Learning, NeurIPS 2025.

[8] Liu et al., Simple and Principled Uncertainty Estimation with Deterministic Deep Learning via Distance Awareness, NeurIPS 2020.

[9] Van Amersfoort et al., Uncertainty Estimation Using a Single Deep Deterministic Neural Network, ICML 2020.

[10] Mukhoti et al., Deep Deterministic Uncertainty: A New Simple Baseline, CVPR 2023.

[11] Gao et al., A Comprehensive Survey on Evidential Deep Learning and Its Applications, IEEE TPAMI 2025.

---

> ### Author Rebuttal · Authors · 2026-03-31
>
> **Q1: Response on the Choice of EDL**
>
> We thank the reviewer for the insightful discussion. We adopted the original EDL form for the following reasons:
>
> **Controlled Validation**. A key contribution of RCA is to link the performance degradation on old-class to a **Confidence Bias**. To rigorously validate this hypothesis, we adopt the most minimal EDL formulation to ensure the gains are attributable to confidence alignment rather than more sophisticated UQ engineering.
>
> **Plug-and-Play Simplicity**. Compared to Bayesian methods with significant computational overhead and architecture-constrained approaches (e.g., SNGP, DUQ), EDL is architecture-agnostic and incurs negligible cost, aligning well with our design goal.
>
> **Extensibility**. RCA is orthogonal to the most improvements of EDL variants, and can be readily combined with them. Exploring such combinations is a promising direction for future work.
>
> **Q2: Generality as a Plug-in Framework**
>
> We integrated RCA into several GCD baselines and report the results below:
>
> | Methods | CUB (All) | CUB (Old) | CUB (New) | FGVC-Aircraft (All) | FGVC-Aircraft (Old) | FGVC-Aircraft (New) |
> |-|-|-|-|-|-|-|
> |ProtoGCD(TPAMI2025)|59.83|65.04|57.22|53.97|61.76|50.07|
> |ProtoGCD+RCA|64.15 |78.52|59.96|59.41|74.37|51.93|
> |SPTNet(ICLR2024)|64.50|66.44|63.53|55.35|57.68|54.18|
> |SPTNet+RCA|66.06|68.91|64.63|59.25|62.18|57.78|
> |MOS(CVPR2025)|67.88|74.45|64.60|62.01|65.73|60.15|
> |MOS+RCA|70.11|78.85|65.73|63.15|72.87|58.29|
>
> Consistent gains across multiple architectures and datasets demonstrate RCA's effectiveness as a plug-and-play module. In addition, ALLGCD's evaluation is precluded by the lack of public code.
>
> We will include these in the revised manuscript.
>
> **Q3: Augmentation Protocol and CCA Design**
>
> **1. Clarification on Augmentation Protocol**
>
> We emphasize that we use standard GCD multi-view augmentation without adding extra views: for each input image, two views $x_i$ and $x'_i$ are generated via standard random augmentation. Our contribution lies in leveraging these existing augmented views to achieve a deeper alignment in the evidential space.
>
> **2.1 Complementarity of first- and second-order alignment**
>
> Geometrically, first-order alignment $L_{1st}$ constrains the centroid (expectation) of the Dirichlet distribution, ensuring that the student matches the teacher’s class probability distribution. However, this is insufficient, as different Dirichlet parameters can share the same centroid while exhibiting significantly different distributional structures.
>
> Second-order alignment $L_{2nd}$ further constrains the "shape" and "concentration" on the simplex, enabling alignment of the underlying uncertainty structure. Therefore, \$L_{2nd}$ is not overly restrictive, but a core mechanism for capturing richer uncertainty structures. Empirically, ablation results show that CCA brings further improvements over RAC alone.
>
> **2.2 Theoretical interpretation**
>
> The necessity of first- and second-order alignment is theoretically supported by our Theorem 4.3. From a PAC-Bayesian perspective, jointly optimizing $L_{1st}$ and $L_{2nd}$ in CCA essentially corresponds to directly optimizing this **theoretical bound**, leading to more complete knowledge transfer.
>
> **Q4: Response On the Uncertainty Quality.**
>
> We thank the reviewer for this insightful suggestion. We address this from two perspectives:
>
> **1. Core Evaluation Metric: Confidence Gap**
>
> As shown in the table below, we compare multiple confidence metrics MSP (Maximum Softmax Probability), Margin (Top-2 Probability Gap), and Negative Entropy on datasets. The results demonstrate that after introducing RCA, the confidence gap between OL (Old Labeled) and OU (Old Unlabeled) is significantly reduced, which is consistent with the motivation (Figure 1,2 in main text).
>
> |Dataset|Metric|Method|OL|OU|OL-OU Confidence Gap|
> |-|-|-|-|-|-|
> |CIFAR100|MSP|Wo/RCA|0.94|0.80|0.14|
> |||W/RCA|0.85|0.81|0.04|
> ||Margin|Wo/RCA|0.94|0.76|0.18|
> |||W/RCA|0.85|0.81|0.04|
> ||Negative Entropy|Wo/RCA|-0.45|-1.17|0.72|
> |||W/RCA|-1.12|-1.31|0.19|
> |CUB|MSP|Wo/RCA|0.83|0.55|0.28|
> |||W/RCA|0.80|0.73|0.07|
> ||Margin|Wo/RCA|0.82|0.50|0.32|
> |||W/RCA|0.79|0.71|0.08|
> ||Negative Entropy|Wo/RCA|-1.22|-2.62|1.40|
> |||W/RCA|-1.53|-1.94|0.41|
>
> **2. Limitations of Calibration Metrics**
>
> We argue that calibration metrics may **not be suitable** for the objective of RCA. For instance, ECE (Expected Calibration Error) measures the correspondence between predicted confidence and actual accuracy, which is distinct from our goal of confidence alignment. Experimental results on the OU data of CIFAR100 illustrate this discrepancy:
>
> -SimGCD: Acc: 78.35|Confidence: 80.71|ECE: 0.0547
>
> -SimGCD + RCA: Acc: 83.18|Confidence: 81.22|ECE: 0.0894
>
> As shown above, SimGCD achieves a lower ECE due to its lower accuracy and more conservative confidence. In contrast, RCA significantly improves classification accuracy, which leads to a higher ECE. We hope this clarification addresses the concern.

---

> > ### Author Rebuttal · Reviewer_oqsm · 2026-04-02
> >
> > I thank the authors for their detailed and comprehensive rebuttal, which addressed most of my concerns. Accordingly, I increase my score to 4 (Weak Accept).

---

> > > ### Author Response · Authors · 2026-04-02
> > >
> > > We sincerely thank the reviewer for the constructive feedback and positive comments, and are glad that our response has addressed your concerns.

---

### Official Review · Reviewer_gQ21 · 2026-03-12

**Soundness:** 3
**Presentation:** 4
**Significance:** 3
**Originality:** 3
**Overall Recommendation:** 4
**Confidence:** 3

**Summary:**

This paper tackles a confidence bias issue in the problem of Generalized Category Discovery (GCD) in which a trained classification model should generalize for seen-class unlabeled & unseen-class unlabeled samples. The authors demonstrate that while the conventional entropy regularization prevents overfitting, it suppresses confidence level across all unlabeled samples, making the seen and unseen class instances undistinguished. To resolve this, the authors propose RCA which leverages the Dirichlet distribution and evidential learning for calibrating confidence. The RAC component is built on the ELBO maximization on labeled data to establish high-fidelity certainty anchors. Meanwhile, a CCA component performs the 1st and 2nd order alignment under a teacher-student framework for all samples. Experiments demonstrate improvements when applied to the current SOTA methods.

**Compliance With Llm Reviewing Policy:**

Affirmed.

**Final Justification:**

The rebuttal mostly addresses my concerns. Following the clarification of the paper's focus and the addition of empirical results, I find the introduction of EDL to GCD to be both novel and effective. Considering the overall contribution, I would maintain my score and lean toward acceptance.

**Key Questions For Authors:**

1. Does the proposed apply to more existing GCD methods?
2. Why is it necessary to align the 1st and 2nd order teacher-student outputs? The paper would benefit from interpreting the theoretical part.
3. What is the crucial key to distinguishing seen and unseen classes under the current theoretical framework?

**Limitations:**

The performance improvement on the new class can be limited and marginal.

**Strengths And Weaknesses:**

Strengths:

The paper identifies a key problem for GCD with clear empirical evidence. The paper is well-written and easy to follow and understand the diagnosis of confidence bias. Meanwhile, the paper is theoretically solid, with rigorous theoretical supports for both components. Empirically, the thorough experimental comparison to existing methods demonstrate the effectiveness of the method.

Weaknesses:

- Does the proposed apply to more existing GCD methods? The current evaluation can be extended.
- The new class improvement is quite marginal on fine-grained datasets.
- From results in table 3&4, the Cross-view Confidence Alignment is not effective or harmful for the new classes.

Minors:
  - $\lambda_t$ already exists in Eq.(7) and should be removed from Eq.(8).
  - The target Dirichlet defined at the end of page 4 seems to be wrong. It should be $(1 - y_i) + y_i \odot \alpha_i$.
  - Typo in Eq.(12).

---

> ### Author Rebuttal · Authors · 2026-03-31
>
> **Q1: Performance of RCA Applied to More GCD Methods.**
>
> We thank the reviewer for constructive suggestions. We conducted **additional experiments by integrating RCA** into several baselines, the results are summarized below:
>
> | Methods | CUB (All) | CUB (Old) | CUB (New) | FGVC-Aircraft (All) | FGVC-Aircraft (Old) | FGVC-Aircraft (New) |
> |-|-|-|-|-|-|-|
> |ProtoGCD(TPAMI2025)|59.83|65.04|57.22|53.97|61.76|50.07|
> |ProtoGCD+RCA|64.15 |78.52|59.96|59.41|74.37|51.93|
> |SPTNet(ICLR2024)|64.50|66.44|63.53|55.35|57.68|54.18|
> |SPTNet+RCA|66.06|68.91|64.63|59.25|62.18|57.78|
> |MOS(CVPR2025)|67.88|74.45|64.60|62.01|65.73|60.15|
> |MOS+RCA|70.11|78.85|65.73|63.15|72.87|58.29|
>
> Consistent improvements across multiple architectures and datasets empirically demonstrate RCA's effectiveness as a plug-and-play module.
>
> We will include these results in the revised manuscript.
>
>
> **Q2: Response On Margin Improvement of New-calss**
>
> **Primary Research Objective**. The primary goal of this work is to establish a clear connection between the degradation of old-class performance and the confidence bias in GCD. Our method is designed to alleviate the inherent stability–plasticity dilemma through proposed RCA, which we consider a challenging and meaningful contribution. As demonstrated in our experiments, RCA brings significant improvements on old classes across multiple benchmarks, validating its effectiveness in addressing this core issue.
>
> **Rationale for New-class Gains**. We do not introduce any dedicated modules specifically tailored to enhance the new-class discovery. Nevertheless, RCA still achieves improvements on new classes in most datasets. We attribute this gain to the calibration of confidence, which leads to clearer decision boundaries. In fine-grained scenarios, further gains on new-class typically rely on stronger self-supervised strategies or specialized attention mechanisms, which beyond the scope of this work. Under our framework, maintaining harmless or improved performance on new-class already meets our intended balance.
>
> **Q3: Response On the necessity of aligning first- and second-order.**
>
> **1. Complementarity of first- and second-order alignment**
>
> Geometrically, first-order alignment $L_{1st}$ constrains the centroid (expectation) of the Dirichlet distribution, ensuring that the student matches the teacher’s class probability distribution. However, this is insufficient, as different Dirichlet parameters can share the same centroid while exhibiting significantly different distributional structures.
>
> Second-order alignment $L_{2nd}$ further constrains the "shape" and "concentration" on the simplex. Therefore, \$L_{2nd}$ is a core mechanism for capturing richer uncertainty structures. The two objectives target different aspects of the Dirichlet representation and are inherently **complementary**. Empirically, ablation results show that CCA brings further improvements over RAC alone.
>
> **2. Theoretical interpretation**
> The necessity of first- and second-order alignment is theoretically supported by our Theorem 4.3. From a PAC-Bayesian perspective, jointly optimizing $L_{1st}$ and $L_{2nd}$ in CCA essentially corresponds to directly optimizing this **theoretical bound**, leading to more complete knowledge transfer.
>
> **Q4: Key to distinguishing seen and unseen.**
>
> In our work, the key to distinguishing seen and unseen classes lies in coordinating their learning strategies. This challenge stems from the asymmetry between the learning mechanisms for old and new classes in GCD, as discussed in the main text with reference to Figures 1 and 2. RCA achieves collaborative optimization between them through confidence alignment. Therefore, the effectiveness of RCA hinges on the degree of confidence alignment, suggesting that future uncertainty-aware methods could be integrated into our framework to further improve performance.
>
> **Q5: Response to Minors.**
> We thank the reviewer for pointing out these minor issues. We will correct them in the final version.
>
> Regarding the $\tilde{\boldsymbol{\alpha}}_i = \mathbf{y}_i + (1 - \mathbf{y}_i) \odot \boldsymbol{\alpha}_i$, we clarify that it is correct. It follows standard practice and is consistent with prior works (e.g., Sensoy et al., 2018).
>
> $KL[D(\mathbf{\mu} \mid \tilde{\boldsymbol{\alpha}}_i) \parallel D(\mathbf{\mu} \mid \mathbf{1})]$ acts as a regularizer to suppress misleading evidence in incorrect classes. By setting the ground-truth class parameter in $\tilde{\boldsymbol{\alpha}}_i$ to $1$, the KL term becomes zero for the correct class, ensuring the model is not penalized for assigning evidence to the true category.

---

> > ### Author Rebuttal · Reviewer_gQ21 · 2026-04-01
> >
> > The authors have partially addressed my concerns. With the clarification of the paper's focus and added empirical results, I believe the paper has some novelty for introducing EDL to GCD, and shows empirical strength in its targeted problem.
> > However, I still could not fully agree on Q2. Separating the old-unlabeled and the new by confidence would purify the new class samples before clustering to facilitate the downstream recognition. What is the distribution of recognized "new" samples compared to the true "new" samples when run with/without the proposed method? Is there any further explanation? This is my final concern.

---

> > > ### Author Response · Authors · 2026-04-02
> > >
> > > **Further Clarification on New-class Performance**
> > >
> > > We thank the reviewer for the insightful follow-up. To better clarify the relatively moderate improvement on new classes, we provide a more detailed error decomposition on CUB and CIFAR100 datasets.
> > >
> > > | CUB | True-New | Pred-New | Correct:New | Err:New-New | Err:New->Old | New ACC (OU+New) | New ACC (Val on New) |
> > > |-|-|-|-|-|-|-|-|
> > > | SimGCD | 2997 | 3183 | 1855 | **920 (30.7%)** | **222 (7.4%)** | 61.90 | 62.23 |
> > > | SimGCD+RCA | 2997 | 3055 | 1853 | 869 (30.0%) | **275 (9.2%)** | 61.66 | **62.50** |
> > >
> > > | CUB | True-Old | Pred-Old | Correct:Old | Err:Old-Old | Err:Old->New | Old ACC (Val on OU+New) | Old ACC (Val on Old) |
> > > |-|-|-|-|-|-|-|-|
> > > | SimGCD | 1499 | 1313 | 1010 | **81 (5.4%)** | **408 (27.2%)** | 67.38 | 71.45 |
> > > | SimGCD+RCA | 1499 | 1441 | 1081 | 85 (5.7%) | **333 (22.2%)** | 72.38 | **74.65** |
> > >
> > > | CIFAR100 | True-New | Pred-New | Correct:New | Err:New-New | Err:New->Old | New ACC (Val on OU+New) | New ACC (Val on New) |
> > > |-|-|-|-|-|-|-|-|
> > > | SimGCD | 10000 | 10937 | 8159 | **1015 (10.1%)** | **826 (8.3%)** | 81.59 | 81.71 |
> > > | SimGCD+RCA | 10000 | 10084 | 8218 | **802 (8.0%)** | **980 (9.8%)** | 82.18 | **82.31** |
> > >
> > > | CIFAR100 | True-Old | Pred-Old | Correct:Old | Err:Old-Old | Err:Old->New | Old ACC (Val on OU+New) | Old ACC (Val on Old) |
> > > |-|-|--|--|-|-|-|-|
> > > | SimGCD | 20000 | 19063 | 15670 | **2567 (12.8%)** | **1763 (8.8%)** | 78.35 | 78.39 |
> > > | SimGCD+RCA | 20000 | 19916 | 16637 | **2299 (11.5%)** | **1064 (5.3%)** | 83.18 | **83.18** |
> > >
> > > **(1) Error decomposition**
> > >
> > > From the results above, we observe distinct behaviors across different datasets:
> > >
> > > CUB (Fine-grained): The primary source of error **for new-class is intra-class confusion (New–New errors)**. In contrast, the main limitation **for old classes lies in Old→New misassignments**, which are significantly reduced by RCA (from 27.2% to 22.2%), leading to substantial improvements in old-class accuracy.
> > >
> > > CIFAR100 (Coarse-grained): Errors are more balanced between inter-class confusion and intra-class misassignment. In this case, RCA brings more consistent improvements to both old and new classes.
> > >
> > > **(2)  Reasons for the marginal improvement of new-class (Especially in fine-grained settings)**
> > >
> > > RCA primarily improves performance through **confidence alignment**, which effectively reduces **Old→New errors**. As pointed out by the reviewer, this leads to a cleaner predicted novel set. Indeed, as shown in the table, the number of predicted novel samples becomes closer to the ground-truth distribution.
> > >
> > > However, from an evaluation perspective, **reducing Old↔New errors does not directly translate into higher new-class accuracy**. This is because the number of correctly classified novel samples (Correct:New) remains largely unchanged on CUB. In fine-grained scenarios, where inter-class differences are subtle, the dominant errors lie within novel classes themselves (New–New confusion).
> > >
> > > To further verify this, we additionally evaluate performance on a clean novel subset (denoted as "New ACC (Val on New)"). The results show: CUB: 62.23 → 62.50. This indicates that RCA does not directly alter the intrinsic separability of new classes, which is primarily governed by representation learning.
> > >
> > > In addition, we observe a slight increase in New→Old errors, suggesting that some difficult samples are reassigned to old classes after confidence alignment, which may slightly limit the improvement on new-class performance.
> > >
> > > Furthermore, in order to further demonstrate the consistency of our motives and methods, we show the **confidence bias** via the gap between labeled and unlabeled old-class samples (OL–OU gap) using MSP (Maximum Softmax Probability), Margin (Top-2 Probability Gap), and Negative Entropy.
> > >
> > > |Dataset|Metric|Method|New|OL (Old Labeled)|OU (Old Unlabeled)|OL-OU Gap|
> > > |-|-|-|-|-|-|-|
> > > |CIFAR100|MSP|Wo/RCA|0.7210|0.9430|0.8029|0.1401|
> > > | | |W/RCA|0.6638|0.8473|0.8113|0.0360|
> > > | |Margin|Wo/RCA|0.6886|0.9364|0.7593|0.1771|
> > > | | |W/RCA|0.6250|0.8451|0.8051|0.0400|
> > > | |Negative Entropy|Wo/RCA|-1.6930|-0.4454|-1.1688|0.7234|
> > > | | |W/RCA|-1.9948|-1.1166|-1.3147|0.1981|
> > > |CUB|MSP|Wo/RCA|0.5591|0.8305|0.5503|0.2802|
> > > | | |W/RCA|0.6402|0.7994|0.7273|0.0721|
> > > | |Margin|Wo/RCA|0.5079|0.8175|0.5007|0.3168|
> > > | | |W/RCA|0.6023|0.7949|0.7129|0.0820|
> > > | |Negative Entropy|Wo/RCA|-2.5517|-1.2212|-2.6169|1.3957|
> > > | | |W/RCA|-2.3539|-1.5266|-1.9396|0.4130|
> > >
> > > As shown in the table above, the proposed RCA establishes a clear connection between confidence bias and the stability–plasticity trade-off in GCD in a plug-and-play manner. By aligning confidence, it effectively mitigates the degradation of old-class performance. This also shifts the remaining challenge toward intra-novel discrimination, suggesting that future work can focus more on improving novel-class separability.

---

### Official Review · Reviewer_4cXi · 2026-03-12

**Soundness:** 3
**Presentation:** 3
**Significance:** 2
**Originality:** 3
**Overall Recommendation:** 4
**Confidence:** 3

**Summary:**

This paper identifies a confidence bias in entropy-regularized GCD, where known-class unlabeled samples are pushed into low-confidence regions, harming stability. The proposed RCA framework uses Evidential Deep Learning to anchor certainty on labeled data via RAC and propagate calibrated uncertainty to unlabeled data via CCA, yielding consistent old-class accuracy improvements across six benchmarks.

**Compliance With Llm Reviewing Policy:**

Affirmed.

**Final Justification:**

Thanks for the rebuttal and the authors' detailed work and explanation. I will raise my rating to weak accept.

**Key Questions For Authors:**

Please see cons

**Limitations:**

yes

**Strengths And Weaknesses:**

Pros

1.The experimental motivation is clearly articulated and the problem analysis is thorough. The paper provides intuitive visualizations of the Confidence Bias phenomenon, offering convincing empirical evidence for the core diagnosis.

2.RCA is integrated into both a weak baseline (SimGCD) and a strong baseline (AFGCD), demonstrating consistent gains across both settings and suggesting that the framework generalizes beyond a specific training recipe.


Cons

1.On CIFAR-10, SimGCD+RCA yields only marginal improvements of 0.1% in All accuracy and 0.6% in Old accuracy (Table 1). While the authors attribute this to the inherent simplicity of the dataset, no quantitative criterion or analysis is provided, leaving readers without a principled way to anticipate under what degree of Confidence Bias RCA can be expected to produce meaningful gains.

2.The parameterization of Dirichlet distributions and the dual-level alignment losses (L1st and L2nd) inevitably introduce additional computational overhead. However, the paper provides no comparison of training time or memory consumption.

3.The experimental setup employs 60 warm-up epochs for RAC and 80 warm-up epochs for CCA (Appendix E.2), hyperparameters that may substantially affect performance. However, the sensitivity analysis presented in Figures 5 and 6 covers only the weighting coefficients λ1 and λ2, with no ablation conducted on the warm-up epoch counts.

---

> ### Author Rebuttal · Authors · 2026-03-31
>
> **Q1: Response to CIFAR10 Marginal Improvement.**
>
> We thank the reviewer for this insightful comment. To provide a quantitative explanation, we analyze the **confidence bias magnitude** via the gap between labeled and unlabeled old-class samples (OL–OU gap) using MSP (Maximum Softmax Probability), Margin (Top-2 Probability Gap), and Negative Entropy.
> |Dataset|Metric|Method|OL (Old Labeled)|OU (Old Unlabeled)|OL-OU Gap|
> |-|-|-|-|-|-|
> |CIFAR10|MSP|Wo/RCA|0.9999|0.9975|0.0024|
> | | |W/RCA|0.9994|0.9981|**0.0013**|
> | |Margin|Wo/RCA|0.9999|0.9955|0.0043|
> | | |W/RCA|0.9991|0.9967|**0.0024**|
> | |Negative Entropy|Wo/RCA|-2.1955|-2.2005|0.0050|
> | | | W/RCA | -2.2162 | -2.2178 | **0.0016** |
> |CIFAR100|MSP|Wo/RCA|0.9430|0.8029|0.1401|
> | | |W/RCA|0.8473|0.8113|**0.0360**|
> | |Margin|Wo/RCA|0.9364|0.7593|0.1771|
> | | |W/RCA|0.8451|0.8051|**0.0400**|
> | |Negative Entropy|Wo/RCA|-0.4454|-1.1688|0.7234|
> | | |W/RCA|-1.1166|-1.3147|**0.1981**|
> |CUB|MSP|Wo/RCA|0.8305|0.5503|0.2802|
> | | |W/RCA|0.7994|0.7273|**0.0721**|
> | |Margin|Wo/RCA|0.8175|0.5007|0.3168|
> | | |W/RCA|0.7949|0.7129|**0.0820**|
> | |Negative Entropy|Wo/RCA|-1.2212|-2.6169|1.3957|
> | | |W/RCA|-1.5266|-1.9396|**0.4130**|
>
> We can observe that:
>
> - CIFAR10 exhibits **a negligible gap (e.g., MSP: 0.0024), indicating minimal confidence bias**.
>
> - CIFAR100 and CUB show substantially larger gaps (e.g., MSP: 0.1401 and 0.2802).
>
> Therefore, since RCA is specifically designed to address confidence bias in GCD, its effectiveness is correlated with the OL–OU confidence gap. This explains the marginal improvement observed on simpler datasets such as CIFAR10, where the gap is minimal, and in turn supports our core motivation. Importantly, when the confidence gap is small, RCA does not degrade performance. Moreover, most methods already achieve near-saturated performance on CIFAR10, further limiting observable gains.
>
> We will include this analysis in the revised version.
>
> **Q2: Response to Concern on Computational Overhead.**
>
> We provide the following clarifications:
> **No additional model structure or parameters**
> RAC and CCA operate solely at the loss level, without introducing extra network modules or learnable parameters. Therefore, the forward pass of the model remains unchanged, and computational cost is still dominated by the backbone.
>
> **Theoretical complexity: lightweight element-wise operations.** From a formulation perspective: RAC (Eq. (6)–(8)) mainly involves digamma/Gamma functions and vector summations; CCA (Eq. (11), (13)) mainly involves KL divergence computations.
> These operations are all O(K) element-wise computations (where K is the number of classes) and do not require additional feature extraction or intermediate representations. Thus, they introduce negligible computational overhead compared to the backbone. In addition, multi-view augmentation is a standard component in GCD. Our method does not introduce additional views or extra backbone forward passes, the additional computation only arises from loss construction at the logits level.
>
> **Empirical training overhead.** We compare the training efficiency in the table below. We can observe that training time per epoch is nearly identical before and after adding RCA:
> | Method | CUB (s/epoch) | CIFAR100 (s/epoch) |
> |-|-|-|
> |SimGCD|16–17|151–154|
> |SimGCD+RCA|16–17|151–154|
> |AFGCD|17|163–165|
> |AFGCD+RCA|17|163–165|
>
> Furthermore, as RCA is only applied during training, no additional overhead is incurred at inference. We will include further clarification on efficiency in the final version.
>
> **Q3:Response to Concern on Warm-up Epoch Sensitivity**
>
> We thank the reviewer for pointing out the warm-up epoch settings. We provide additional clarification:
>
> 1. **Warm-up in GCD methods**
> Most GCD methods employ self-distillation strategies, where the teacher (from the previous epoch) is warmed up for ~30 epochs. Following this common practice, RAC is also warmed up for 30 epochs after the teacher warm-up. This ensures stable initial optimization while keeping the implementation simple.
>
> 2. **Sensitivity of CCA to warm-up**
> We conducted an ablation on the number of warm-up epochs for CCA. The results are summarized below:
>
> |Dataset / Epoch|60|80|100|120|
> |-|-|-|-|-|
> |CUB|71.11|71.20|71.15|71.17|
> |CIFAR100|84.72|84.76|84.65|84.65|
>
> As shown, CCA provides stable gains across most warm-up settings, demonstrating its robustness to this hyperparameter.

---

> > ### Author Rebuttal · Reviewer_4cXi · 2026-04-05
> >
> > Thanks for the rebuttal and the authors' detailed work and explanation. I will raise my rating to weak accept.

---

> > > ### Author Response · Authors · 2026-04-06
> > >
> > > We are grateful to reviewer for the insightful suggestions and are glad that our response has effectively addressed your concerns.

---

### Official Review · Reviewer_7tQt · 2026-03-13

**Soundness:** 3
**Presentation:** 2
**Significance:** 2
**Originality:** 2
**Overall Recommendation:** 4
**Confidence:** 4

**Summary:**

This paper studies the problem of Generalized Category Discovery and identifies what the authors call a confidence bias in existing parametric approaches caused by entropy regularization on unlabeled data. The paper argues that this bias suppresses predictive certainty for unlabeled samples and leads to a mismatch between labeled and unlabeled instances even when they belong to the same category. To address this issue, the authors propose Reliable Confidence Alignment, a plug-and-play framework based on evidential deep learning. The framework introduces two key components: Reliable Anchor for Certainty to establish high-confidence anchors using labeled samples, and Cross-view Confidence Alignment to propagate reliable confidence to unlabeled data through multi-view consistency.

**Compliance With Llm Reviewing Policy:**

Affirmed.

**Final Justification:**

I thank the authors for their detailed and comprehensive rebuttal, which addressed most of my concerns. Accordingly, I increase my score to 4 (Weak Accept).

**Key Questions For Authors:**

1.	The RAC and CCA modules appear to introduce multiple sources of regularization. Which component contributes most to the final performance gain? A more detailed ablation isolating evidential modeling from cross-view alignment would help clarify this.
2.	Does the method remain stable when the proportion of novel classes increases significantly? How sensitive is RCA to the labeled/unlabeled class ratio?

**Limitations:**

yes

**Strengths And Weaknesses:**

**Strengths:**

1.	The proposed framework is presented as a plug-and-play component that could potentially be integrated into existing parametric GCD pipelines.
2.	The method is tested on both coarse-grained and fine-grained datasets, which is appropriate for validating robustness across different semantic granularity levels.

**Weakness:**

1.	The paper asserts that entropy regularization “indiscriminately suppresses predictive certainty,” creating a distributional wedge between labeled and unlabeled data. However, this claim is largely intuitive and descriptive, lacking rigorous theoretical analysis or convincing empirical diagnostics. The work does not provide a formal definition or quantitative metric of this supposed bias.
2.	The RAC and CCA modules are described at a relatively high level, but their mechanistic contribution is not deeply analyzed. It remains unclear whether the observed improvements stem from the evidential formulation itself, from confidence reweighting, or from additional regularization introduced by multi-view consistency.

---

> ### Author Rebuttal · Authors · 2026-03-31
>
> **Q1:Further clarification of Confidence Bias.**
>
> We appreciate the reviewer’s request for a more rigorous foundation. To clarify that our identification is supported by empirical analysis, we provide:
>
> **1.Theoretical Analysis of Gradient Conflict Caused by Entropy Regularization (ER)**
>
> The gradient of ER for logit $z_{i,k}$ is:
> $$\frac{\partial L_{\mathrm{ER}}}{\partial z_{i,k}} = \frac{p_i^{(k)}}{N} \left( \log \bar{p}^{(k)} - \sum_j p_i^{(j)} \log \bar{p}^{(j)} \right)$$
>
> While labeled data is driven toward low entropy state ($H \to 0$) by Cross-Entropy (CE), unlabeled data is only subject to the smoothing force of ER loss to prevent class collapse. This asymmetry causes unlabeled samples to remain in a lower-confidence state.
>
> To counter this, RCA introduces a compensatory gradient, shifts from volatile Softmax point estimates to Dirichlet-based evidence modeling: Taking the RAC as an example, its objective consists of two terms. $L_{\mathrm{risk}}$: Accumulates absolute evidence $S_i$ (high-belief state $b_k$) for labeled classes, making them robust against ER’s smoothing pressure. $L_{\mathrm{reg}}$: Penalizes misleading evidence to maintain high uncertainty on non-targets, preventing overconfidence. The CCA extends this mechanism to the unlabeled data.
>
> By leveraging cumulative Dirichlet mass rather than volatile Softmax scores, RCA provides "confidence compensation" that effectively closes the wedge.
>
> **2.Formal Definition: Confidence Gap**
>
> We define the **Confidence Gap ($\Delta C$)** as the discrepancy in predictive certainty $C(p)$ between labeled  and unlabeled instances of the same category $k$:
>
> $\Delta C_k = E_{x \in D_{l,k}} [C(p)] - E_{x \in D_{u,k}} [C(p)]$
>
> $C(p)$ is measured by: MSP ($\max p^{(i)}$), Margin ($p_{max} - p_{2nd}$), and Negative Entropy ($-H(\mathbf{p})$).
>
> **3.Empirical Diagnostics and Manuscript Evidence**
>
> Results presented below confirm that traditional methods (SimGCD) exhibit a severe $\Delta C$, whereas RCA minimizes this gap across all metrics.
> |Dataset|Metric|Method|OL (Old Labeled)|OU (Old Unlabeled)|OL-OU Gap|
> |-|-|-|-|-|-|
> |CIFAR100|MSP|Wo/RCA|0.94|0.80|0.14|
> | | |W/RCA|0.85|0.81|**0.04**|
> | |Margin|Wo/RCA|0.94|0.76|0.18|
> | | |W/RCA|0.85|0.81|**0.04**|
> | |Negative Entropy|Wo/RCA|-0.45|-1.17|0.72|
> | | |W/RCA|-1.12|-1.31|**0.19**|
> |CUB|MSP|Wo/RCA|0.83|0.55|0.28|
> | | |W/RCA|0.80|0.73|**0.07**|
> | |Margin|Wo/RCA|0.82|0.50|0.32|
> | | |W/RCA|0.79|0.71|**0.08**|
> | |Negative Entropy|Wo/RCA|-1.22|-2.62|1.40|
> | | |W/RCA|-1.53|-1.94|**0.41**|
>
> **Q2:Response on Mechanism Contribution and Component Roles.**
>
> **1.Relationship between RAC and CCA**
>
> RAC and CCA are not parallel or redundant modules, but instead form a **coupled mechanism with a clear dependency structure**. RAC first transforms the model into an evidential network, learning Dirichlet parameters to establish reliable high-confidence anchors. Based on this, CCA performs cross-view alignment over the confidence structure of unlabeled data. Without RAC, the model only produces softmax-based point estimates. In this case, the Dirichlet distributions that CCA relies on lack semantic grounding and cannot be meaningfully defined.
>
> **2.Ablation and component contribution**
>
> Ablation (Tables 3–4) confirms RAC is the primary contributor, significantly boosting old-class accuracy and mitigating confidence bias. CCA provides additive gains by propagating the learned confidence structure to unlabeled data, a trend consistent across all baselines.
>
> **3.On "Additional regularization from multi-view consistency"**
>
> We emphasize that we use standard GCD multi-view augmentation without adding extra views. Our performance gains stem from a more expressive alignment objective rather than additional regularization.
>
> **Q3:Response on Sensitivity to Novel-Class Proportion.**
>
> To further clarify the robustness of RCA, we conducted the following experiment:
>
> **Varying the proportion of labeled data**. We set the labeled data ratio to 0.25/0.375/0.5, and report the results below:
> |Ratio|Method|All|Old|New|
> |-|-|-|-|-|
> |0.25|SimGCD|53.8|53.7|53.8|
> |-|+RCA|55.6|56.9|54.7|
> |0.375|SimGCD|59.1|61.2|57.8|
> |-|+RCA|60.3|64.4|57.8|
> |0.5|SimGCD|63.7|67.4|61.9|
> |-|+RCA|65.2|72.4|61.7|
>
> **Varying the number of new-class**. We set the labeled classes number to 60/80/100, thereby increasing the proportion of novel classes:
> |Labeled Classes|Method|All|Old|New|
> |-|-|-|-|-|
> |60|SimGCD|52.3|41.4|54.6|
> |-|+RCA|54.8|**54.7**|54.8|
> |80|SimGCD|60.6|54.9|62.6|
> |-|+RCA|62.5|**61.0**|62.9|
> |100|SimGCD|63.7|67.4|61.9|
> |-|+RCA|65.2|**72.4**|61.7|
>
> These results confirm RCA's **robustness under challenging settings**. Notably, when the novel-class number increases, the negative effect of ER becomes more pronounced. This observation further supports our core motivation and highlights the effectiveness of RCA in mitigating confidence bias.

---

> > ### Author Rebuttal · Reviewer_7tQt · 2026-04-04
> >
> > Thank you for the detailed and constructive rebuttal.
> >
> > However, some concerns remain only partially addressed.
> >
> > - What is the confidence gap for novel classes? Why do the metrics on old-labeled decrease significantly as Table 1 in the rebuttal?
> >
> > - How do you perform the experiments about sensitivity to novel-class proportion? The performance seems to be not stable when the novel-class number increases, such as 54.8 for 60 and 62.9 for 80. What will happen when setting the number of labelled classes to 20 or 10?
> >
> > Thus, I will keep my current score.

---

> > > ### Author Response · Authors · 2026-04-05
> > >
> > > **Comprehensive Demonstration of Confidence Levels on Old and New Classes**
> > >
> > > The table below presents a detailed comparison between the baseline method (SimGCD) and the proposed RCA under two representative scenarios: CIFAR-100 (coarse-grained) and CUB (fine-grained).
> > >
> > > |Dataset|Metric|Method|New|OL (Old Labeled)|OU (Old Unlabeled)|OL-OU Gap|
> > > |-|-|-|-|-|-|-|
> > > |CIFAR100|MSP|Wo/RCA|0.7210|0.9430|0.8029|0.1401|
> > > | | |W/RCA|0.6638|0.8473|0.8113|0.0360|
> > > | |Margin|Wo/RCA|0.6886|0.9364|0.7593|0.1771|
> > > | | |W/RCA|0.6250|0.8451|0.8051|0.0400|
> > > | |Negative Entropy|Wo/RCA|-1.6930|-0.4454|-1.1688|0.7234|
> > > | | |W/RCA|-1.9948|-1.1166|-1.3147|0.1981|
> > > |CUB|MSP|Wo/RCA|0.5591|0.8305|0.5503|0.2802|
> > > | | |W/RCA|0.6402|0.7994|0.7273|0.0721|
> > > | |Margin|Wo/RCA|0.5079|0.8175|0.5007|0.3168|
> > > | | |W/RCA|0.6023|0.7949|0.7129|0.0820|
> > > | |Negative Entropy|Wo/RCA|-2.5517|-1.2212|-2.6169|1.3957|
> > > | | |W/RCA|-2.3539|-1.5266|-1.9396|0.4130|
> > >
> > > As discussed in the Introduction (referring to Figure 1) and Section 3.2 (referring to Figure 2) of the main paper, our empirical analysis establishes a clear connection between confidence bias and the stability–plasticity trade-off in GCD. The proposed RCA effectively mitigates the performance degradation on old classes through confidence alignment. This efficient alignment benefits from two aspects. Take the main component RAC as an example: First, the RAC module compensates the confidence of unlabeled old-class data by anchoring the confidence of labeled data, with the detailed rationale and analysis provided in the main paper and our response to Q1 (regarding $L_{\mathrm{risk}}$). Second, $L_{\mathrm{reg}}$ penalizes misleading evidence to maintain high uncertainty on non‑targets, **preventing overconfidence**. Consequently, $L_{\mathrm{reg}}$ causes the observed decrease in confidence scores for Old‑Labeled data. **This is not a harmful operation but a necessary strategy for achieving further alignment.**
> > >
> > > Moreover, for new classes, due to the lack of supervisory signals, their confidence should remain relatively low (compared to old classes). As can be seen from the table, the impact of RCA on the confidence of new classes is **reasonable**.
> > >
> > >
> > > **Regarding Extreme Settings (e.g., Only 10 or 20 Labeled Classes)**
> > >
> > > It should be noted that configurations with only 10 or 20 labeled classes are **far beyond the current GCD setting range** (which typically assumes that labeled classes account for about half). When labeled data are extremely scarce, the task becomes closer to unsupervised clustering. In the main paper, we adopt the standard class split settings, which ensures fair comparison.
> > >
> > > Nevertheless, RCA still achieves excellent performance under highly challenging settings (with very few old classes), as demonstrated in our response to Q3 and the results on CUB (contains 200 classes) under even more demanding settings (see table below).
> > >
> > > |Labeled Classes|Method|All|Old|New|
> > > |-|-|-|-|-|
> > > |10|SimGCD|32.7|20.3|33.4|
> > > |-|+RCA|**34.1**|**25.3**|**34.6**|
> > > |20|SimGCD|28.1|10.7|28.6|
> > > |-|+RCA|**29.0**|**20.7**|**29.2**|
> > >
> > > Moreover, performance fluctuations when the number of novel classes increases are **entirely normal** and arise from **multiple factors**, such as the intrinsic difficulty of the task varying with class composition, and the different balance between old and new classes affecting classification difficulty.
> > >
> > > Importantly, across all settings, RCA consistently improves the accuracy on old classes while hardly harming—and sometimes even benefiting—the performance on new classes. This indicates that we have achieved a sufficient balance between stability and plasticity, further validating the **consistency of our motivation and method**.
> > >
> > > We sincerely hope that this response has adequately addressed your concerns.

---

### Decision · Program_Chairs · 2026-04-30

**Decision:**

Accept (regular)

**Comment:**

This paper studies Generalized Category Discovery (GCD) and identifies a confidence bias in existing parametric methods: entropy regularization reduces certainty on all unlabeled samples, which can harm stability on known classes while the model attempts to discover novel ones. To address this issue, the paper proposes Reliable Confidence Alignment (RCA), a plug-and-play framework based on evidential deep learning that establishes reliable certainty anchors on labeled data and transfers this grounded reliability to unlabeled data through cross-view alignment. Overall, the proposed method improves uncertainty calibration and helps preserve performance on known classes without sacrificing novel-class discovery.

The proposed approach is well motivated and technically sound. The authors provided a detailed and convincing rebuttal that clarified several important concerns raised by the reviewers. These clarifications, together with the additional results in the rebuttal, strengthened the overall presentation and support for the paper. As a result, three reviewers raised their scores, while the remaining reviewer maintained a positive assessment.

Overall, the paper makes a solid contribution to the GCD literature. For the final version, the authors are encouraged to carefully incorporate the reviewers’ comments and suggestions to further strengthen the paper.